

# Long-term memory magnetic correlations in the Hubbard model: A dynamical mean-field theory analysis

Clemens Watzenböck[⋆], Martina Fellinger, Karsten Held and Alessandro Toschi[†]

Institute of Solid State Physics, TU Wien, 1040 Vienna, Austria

⋆ clemens.watzenboeck@tuwien.ac.at , † toschi@ifp.tuwien.ac.at

## Abstract

We investigate the onset of a *not-decaying* asymptotic behavior of temporal magnetic correlations in the Hubbard model in infinite dimensions. This long-term memory feature of dynamical spin correlations can be precisely quantified by computing the difference between the zero-frequency limit of the Kubo susceptibility and the corresponding static isothermal one. Here, we present a procedure for reliably evaluating this difference starting from imaginary time-axis data, and apply it to the testbed case of the Mott-Hubbard metal-insulator transition (MIT). At low temperatures, we find long-term memory effects in the entire Mott regime, abruptly ending at the first order MIT. This directly reflects the underlying local moment physics and the associated *degeneracy* in the many-electron spectrum. At higher temperatures, a more gradual onset of an infinitely-long time-decay of magnetic correlations occurs in the crossover regime, not too far from the Widom line emerging from the critical point of the MIT. Our work has relevant algorithmic implications for the analytical continuation of dynamical susceptibilities in strongly correlated regimes and offers a new perspective for unveiling fundamental properties of the many-particle spectrum of the problem under scrutiny.



# 1 Introduction

The emergence of time/energy scales of different orders of magnitude often represents a pivotal aspect in shaping the physics of many-particle systems. This is certainly the case for the textbook situation of electronic and lattice degrees of freedom in standard materials: The mass difference between electrons and nuclei is directly reflected in the different timescales of the respective dynamics, which underlie the widespread applicability of the adiabatic approximation in solid state physics and of the Migdal theorem for conventional superconductors. Another relevant example is provided by the slowing-down of quantum critical fluctuations, which is intrinsically associated to the occurrence of second-order (quantum) phase transitions.

In the case of correlated systems, the repeated scattering process between electrons represents an additional, but not less important, source of differentiation for the timescales of the relevant fluctuations. For instance, the proximity to a Mott-Hubbard [1,2] or to a Hund's-Mott [3,4] metal-insulator transition (MIT) is known to be associated with a progressive slowing-down [5,6] of the local spin fluctuations, heralding the formation of local magnetic moments in the Mott insulating phases. Generally, one expects a slowing-down of fluctuations to be reflected in a softening [7] of the corresponding peak in the absorption spectra. A similar evolution of the lowest energy absorption peak has been identified [8,9] in the spectral functions describing orbital fluctuations in systems with an effective local attraction. Indeed, as it has been recently pointed out [10], a correlation-driven dilatation of characteristic timescales can have a significant impact onto spectroscopic measurements of correlated systems, such as, e.g., the iron pnictides/chalcogenides. An accurate estimation of the size of the magnetic

moment on their Fe atoms by means of inelastic neutron scattering experiments becomes only possible in the most correlated families of this class of materials [10], when a sufficiently strong slowing-down of local magnetic fluctuations makes the corresponding timescale larger [10–13] than the characteristic one of the neutron spectroscopy probe.

Generally speaking, the significant slowing-down of local spin correlations associated with strong electronic scattering may play an important role for the applicability of the adiabatic spin dynamics (ASD) approach, recently used [14] to investigate anomalous precession effects in semiclassical spin-fermion models, to correlated quantum matter systems. Further, it also yields the necessary timescale separation underlying the insightful description *á la* Landau-Ginzburg of the local moment formation in the Hubbard model, which has been recently proposed in Ref. [15].

By sufficiently large interaction values, however, it is conceivable that a *qualitative* change in the asymptotic behavior of temporal correlations may occur on top of this typical trend [16, 17]: Slowing down effects could become so pronounced that a complete decay of the corresponding fluctuations is no longer observed even for $t \rightarrow \infty$. From a spectroscopic perspective, this would correspond to a complete softening of the lowest absorption peak, eventually collapsing into a $\delta$-function with zero weight at zero frequency [7].

Heuristically, one could associate an asymptotically not decaying part of a given (e.g., magnetic) correlation function to the persistence of "long-term memory" information in the fluctuations of the corresponding sector. On a more formal level, as already noted in some of the earliest works on the linear response theory [18, 19], such a long-term memory behavior of temporal fluctuations precisely reflects the existence of a *finite* difference ($C \neq 0$) between the values of the static *isothermal* susceptibility and the zero-frequency limit of the standard *Kubo response function* in the same system [18, 20]. At the same time, it is important to stress how long-term memory effects are directly linked to intrinsic properties of the many-particle energy spectrum of the problem under consideration. For example, under certain conditions (discussed in Sec. II), a non vanishing $C$ can be associated with the presence of (at least) one *degenerate* many-electron eigenstate. Hence, studying the possible appearance of a long-term memory behavior of temporal correlations in actual model calculations might unveil fundamental information about the many-electron system under investigation.

Eventually, it is also clear that the presence of finite values of $C$ must be taken carefully into account also from an algorithmic perspective. This is certainly the case, for instance, when extracting physical information from a many-body calculation via post-processing approaches like analytic continuation or fluctuation diagnostic schemes.

In this paper, we will first concisely summarize the multi-faceted formal aspects underlying the onset of a long-term memory behavior of temporal correlations and of their mutual interconnections, as addressed in a collection of different literature works. Starting from these considerations, we will illustrate a specific procedure to quantitatively estimate the value of a non-vanishing $C$ from many-body correlation functions computed on the imaginary time. Eventually, by exploiting this procedure we will investigate an emerging long-term memory behavior of the local magnetic fluctuations in the phase diagram of the half-filled Hubbard model solved by means of dynamical mean-field theory (DMFT). In particular, we will discuss the relation of long-term memory effects with the Mott-Hubbard metal insulator transition described by DMFT, and with the associated crossover regime at higher temperature. Finally, relying on a critical analysis of the results obtained, we outline relevant physical and algorithmic implications of the possible emergence of long-term memory effects in many electron systems.

## 2 Theoretical Background

### 2.1 General relationship between measurements and long-time decay

Starting point for our considerations is the intrinsic difference between (i) the zero-frequency limit of the standard Kubo (or: *isolated*) susceptibility $\chi^{\mathcal{R}}(\omega = 0^+)$, (ii) the static *isothermal* susceptibility $\chi^T$, and (iii) the isentropic susceptibility $\chi^S$. These functions define the linear response to an external perturbation in three distinct experimental set-ups: (i) for a system which is (or can be assumed to be) *isolated* from its environment during the action of the external perturbation, (ii) for a system in *thermal* equilibrium with its environment, (iii) for a system *adiabatically* isolated (i.e., kept at fixed entropy $S$), respectively.

In practice, the set-up (i) can be realized by considering dynamical perturbations whose characteristic timescales are *faster* than the thermalization processes between the probe and the environment. Formally, this allows for the application of the conventional Kubo-Nakano response theory [18, 21–23], whose derivation relies on the von Neumann equation for the time-dependent density matrix of the perturbed system:

$$\hat{\rho}_t = \sum_N \frac{e^{-\beta E_N}}{Z} |\Psi_N(t)\rangle\langle\Psi_N(t)|, \tag{1}$$

where $E_N$ and $Z$ are the eigenenergies and the partition function of the unperturbed Hamiltonian $H$, while $|\Psi_N(t)\rangle$ represents the time-evolved $N$−th eigenstate *in the presence* of the external perturbation $\mathcal{F}(t)$ [21]. This expression implicitly implies the isolation of the system during the action of the perturbation, as the Maxwell-Boltzmann weights of $\hat{\rho}_t$ remain *frozen* to the corresponding values of the *unperturbed* system.

The set-ups (ii) and (iii), instead, can be directly linked to the application of a *purely static* perturbation $\mathcal{F}(t) = $ const. (e.g., a static magnetic field). One thus has: $\hat{\mathcal{H}} = \hat{H} - \mathcal{F}\hat{A}$, where $\hat{A}$ is an observable of the system directly coupled with the external perturbation, while the linear susceptibilities associated to measurement of a generic system observable $\hat{B}$ is given by [19]:

$$
\begin{aligned}
\chi^T &= \left.\frac{\partial \langle B\rangle_{\mathcal{F}}}{\partial \mathcal{F}}\right|_T \quad \text{and} \quad \chi^S = \left.\frac{\partial \langle B\rangle_{\mathcal{F}}}{\partial \mathcal{F}}\right|_S \quad \text{where} \\
\langle\hat{B}\rangle_{\mathcal{F}} &= \text{Tr}[\hat{\rho}_{\mathcal{H}}\hat{B}] \quad \text{and} \quad \hat{\rho}_{\mathcal{H}} = \exp(-\beta\hat{\mathcal{H}})/\text{Tr}[\exp(-\beta\hat{\mathcal{H}})].
\end{aligned}
\tag{2}
$$

The Maxwell-Boltzmann weights to be considered in this case are those of the (statically) *perturbed* Hamiltonian, which evidently corresponds to the assumption of a *full thermalization* of the system *in the presence* of the perturbation.

Not surprisingly, the fundamental distinction between these setups may reflect in different values of the corresponding susceptibilities at zero-frequency. In particular, though not often mentioned in the most recent literature, it is known from the earliest works [18] on linear response theory (LRT) and rigorously shown by Wilcox [19], that the zero-frequency limit of the dynamical Kubo susceptibility $\chi^{\mathcal{R}}(\omega = 0)$ is bound from above by the isentropic susceptibility $\chi^S$, which in turn is bound from above by the static isothermal susceptibility $\chi^T$:

$$\chi^T \geq \chi^S \geq \chi^{\mathcal{R}}(\omega = 0). \tag{3}$$

The difference between the isothermal and the zero-frequency limit of the Kubo susceptibility can be also regarded as a measure of the (non-) ergodicity. In particular, one can show (see [18, 24]) that for two generic hermitian operators $\hat{A}$ and $\hat{B}$

$$
\begin{aligned}
\chi^T - \chi^{\mathcal{R}}(\omega = 0) &= \beta \lim_{t\to\infty} \langle\hat{B}\hat{A}(t)\rangle - \beta \langle A\rangle\langle B\rangle \\
&= \beta \lim_{\mathcal{T}\to\infty} \frac{1}{\mathcal{T}} \int_0^{\mathcal{T}} \langle\Delta B\Delta A(t)\rangle \, dt,
\end{aligned}
\tag{4}
$$

where $\Delta \hat{A} = \hat{A} - \langle \hat{A} \rangle$. Further we employ the usual notation for the thermodynamic average $\langle ... \rangle = \frac{1}{Z} \text{Tr} e^{-\beta \hat{H}} ...$, $Z = \text{Tr} e^{-\beta \hat{H}}$, $\beta = \frac{1}{T}$ is the inverse temperature ($k_B = 1$), and $\hat{H}$ the (unperturbed) Hamiltonian. For convenience we define the difference between the two susceptibilities as $\beta C_+$.

$$\beta C_+ := \chi^T - \chi^{\mathcal{R}}(\omega = 0). \tag{5}$$

The analogous constant for negative times is referred to as $C_-$

$$
\begin{aligned}
C_+ &= \lim_{t \to +\infty} \langle \hat{B}\hat{A}(t) \rangle - \langle A \rangle \langle B \rangle \,, \\
C_- &= \lim_{t \to -\infty} \langle \hat{B}\hat{A}(t) \rangle - \langle A \rangle \langle B \rangle \,.
\end{aligned} \tag{6}
$$

Evidently, the equation above highlights the direct link between the (infinitely) long-term correlations (between $A$ and $B$) and the discrepancies between the static isothermal susceptibility and the zero-frequency limit of the dynamical Kubo response. Note that in general $C_+$ and $C_-$ do not have to be equal, also differing from their averaged value:

$$C := \frac{1}{2}(C_+ + C_-). \tag{7}$$

However, Kwok [20] showed that if the (unperturbed) Hamiltonian is invariant under time-reversal and additionally the operators $\hat{A}$ and $\hat{B}$ have the same sign under time-reversal [1] the two constants are the same

$$C_+ = C_- = C. \tag{8}$$

Clearly, this applies to the case we consider in our work: the spin-spin susceptibility ($\hat{A} = \hat{B} = \hat{S_z}$) in the Hubbard model.

As we discuss later, the precise value of $C$ is intimately related to the symmetries of the many-body system under investigation and to the relation of the specific observable $\hat{A}\hat{B}$ considered to that symmetries. In this respect, it is not surprising that the determination of $C$ might encode insightful information about the many-particle energy spectrum of the problem studied, e.g. about the presence of degeneracies.

Eventually, for the sake of clarity, we will assume without loss of generality that $\langle \hat{B} \rangle = 0$. This choice is fully in the spirit of the LRT, as we will explicitly consider the fluctuations with respect to the equilibrium value.

## 2.2 Formalization of the problem

### 2.2.1 Susceptibilities and correlation functions

Several related susceptibilities and correlation functions are commonly used in the context of the LRT. For the generic hermitian operators $\hat{A}$ and $\hat{B}$ they are defined as:

$$
\begin{aligned}
\chi^c_{AB}(t) &= \langle [\hat{A}(t), \hat{B}(0)] \rangle \,, \\
\Psi_{AB}(t) &= i \langle \{\hat{A}(t), \hat{B}(0)\} \rangle \,, \\
\chi^{\mathcal{R}}_{AB}(t) &= i\theta(t) \langle [\hat{A}(t), \hat{B}(0)] \rangle \,, \\
\chi^{\mathcal{A}}_{AB}(t) &= -i\theta(-t) \langle [\hat{A}(t), \hat{B}(0)] \rangle \,,
\end{aligned} \tag{9}
$$

where the operators are expressed in the Heisenberg representation $\hat{A}(t) = e^{i\hat{H}t} \hat{A} e^{-i\hat{H}t}$. The last two expressions, $\chi^{\mathcal{R}}$ and $\chi^{\mathcal{A}}$, correspond to the well-known definition of the retarded and advanced susceptibilities in the Kubo formalism, respectively. All of them can be formally rewritten in terms of two fundamental "building blocks":

$$
\begin{aligned}
\chi^<_{AB}(t) &= i \langle \hat{B}(0)\hat{A}(t) \rangle \,, \\
\chi^>_{AB}(t) &= i \langle \hat{A}(t)\hat{B}(0) \rangle \,.
\end{aligned} \tag{10}
$$

---

[1] in the sense that $T\hat{A}T^{-1} = \epsilon_A \hat{A}$; $T\hat{B}T^{-1} = \epsilon_B \hat{B}$ and $\epsilon_A \epsilon_B = +1$.

For the theoretical background of our paper it is instructive to illustrate their general properties and connections, see e.g. [20]. To this end, we resort to the Lehmann representation [23, 25], where the lesser and the greater susceptibility have the following spectral representation:

$$
\begin{aligned}
\chi_{AB}^{<}(t) &= i\sum_{n,m} e^{-\beta E_n} B_{nm} A_{mn} e^{iE_{mn}t}, \\
\chi_{AB}^{>}(t) &= i\sum_{n,m} e^{-\beta E_m} B_{nm} A_{mn} e^{iE_{mn}t},
\end{aligned}
\tag{11}
$$

where $|E_n\rangle$ are the eigenstates of the Hamiltonian ($\hat{H}|E_n\rangle = E_n|E_n\rangle$), $E_{nm} := E_n - E_m$ denotes the difference between eigenenergies, and $A_{nm} := \langle E_n|\hat{A}|E_m\rangle$ the matrix-elements of $\hat{A}$ in the eigenbasis of $\hat{H}$.

For the following formal derivation, it is useful to consider the time $t$ as a complex variable ($\Re t + i\Im t$). From the exponents in Eq. (11) it is clear that since the spectrum of $\hat{H}$ is bound from below, but not necessarily from above, the corresponding region of analyticity is

$$
\begin{aligned}
\chi_{AB}^{<}(t): &\quad 0 \le \quad \Im t \quad < \beta, \\
\chi_{AB}^{>}(t): &\quad -\beta < \quad \Im t \quad \le 0.
\end{aligned}
\tag{12}
$$

From Eq. (11) it is also evident that

$$
\chi_{AB}^{<}(t+i\beta) = \chi_{AB}^{>}(t),
\tag{13}
$$

or equivalently for the Fourier transform

$$
\chi_{AB}^{>}(\omega) = \chi_{AB}^{<}(\omega)e^{\beta\omega}.
\tag{14}
$$

This allows to express the commutator Green's function as

$$
\chi_{AB}^{c}(\omega) = \frac{1}{i}(e^{\beta\omega}-1)\chi_{AB}^{<}(\omega).
\tag{15}
$$

However, one should be careful when inverting Eq. (15) to express $\chi_{AB}^{<}(\omega)$ in terms of $\chi_{AB}^{c}(\omega)$. As pointed out by [20], there is no guarantee that the Fourier-transform of $\chi^{<}$ converges at $\omega = 0$. This possible problem can however be circumvented by explicitly treating the long-term asymptotic of $\chi^{<}(t)$: $\lim_{t\to\pm\infty}\chi^{<}(t) = iC_{+/-}$. Subtracting this asymptotics, we obtain a quantity which is guaranteed to have a well behaved Fourier transform in the vicinity of $\omega = 0$:

$$
\tilde{\chi}^{<}(t) := \chi^{<}(t) - i\theta(t)C_+ - i\theta(-t)C_-.
\tag{16}
$$

Its Fourier transform is then given by

$$
\begin{aligned}
\tilde{\chi}^{<}(\omega) &= \chi^{<}(\omega) + \frac{C_+}{\omega+i0^+} - \frac{C_-}{\omega-i0^-} \\
&= \chi^{<}(\omega) + (C_+ - C_-)\mathcal{P}\frac{1}{\omega} - 2\pi i\delta(\omega)C,
\end{aligned}
\tag{17}
$$

where $\mathcal{P}$ denotes the Cauchy principal value and $C$ is defined in Eq. (7). Inserting Eq. (17) for $\chi^{<}(\omega)$ into Eq. (15) gives[2]

$$
\chi^{c}(\omega) = \frac{1}{i}(e^{\beta\omega}-1)\left(\tilde{\chi}^{<}(\omega) - (C_+ - C_-)\mathcal{P}\frac{1}{\omega}\right).
\tag{18}
$$

From Eq. (18) we see that also the commutator susceptibility is well behaved at $\omega = 0$ and equals $\chi^{c}(\omega=0) = i\beta(C_+ - C_-)$. Solving Eq. (18) for $\tilde{\chi}^{<}(\omega)$ gives

$$
\tilde{\chi}^{<}(\omega) = i\mathcal{P}\frac{\chi^{c}(\omega)}{e^{\beta\omega}-1} + (C_+ - C_-)\mathcal{P}\frac{1}{\omega},
\tag{19}
$$

---

[2]The $\delta(\omega)$ term in Eq. (17) does not contribute to Eq. (18) because $e^{\beta\omega}-1=0$ for $\omega=0$. See also Appendix C for an alternative derivation where this term $\propto \omega\delta(\omega)$ is treated in a limit procedure. This leads to the same final results.

Inserting Eq. (19) into Eq. (17) gives the desired inverse relationship of Eq. (15):

$$
\begin{aligned}
\chi^{<}(\omega) &= 2\pi i C \delta(\omega) + i\mathcal{P}\frac{\chi^c(\omega)}{e^{\beta\omega}-1}, \\
\chi^{<}(t) &= iC + i\frac{1}{2\pi}\mathcal{P}\int_{-\infty}^{\infty}\frac{\chi^c(\omega)}{e^{\beta\omega}-1}e^{-i\omega t}d\omega.
\end{aligned}
\tag{20}
$$

Equation (14) together with Eq. (20) also establishes:

$$
\begin{aligned}
\chi^{>}(\omega) &= 2\pi i C \delta(\omega) + i\mathcal{P}\frac{\chi^c(\omega)e^{\beta\omega}}{e^{\beta\omega}-1}, \\
\chi^{>}(t) &= iC + i\frac{1}{2\pi}\mathcal{P}\int_{-\infty}^{\infty}\frac{\chi^c(\omega)e^{\beta\omega}}{e^{\beta\omega}-1}e^{-i\omega t}d\omega.
\end{aligned}
\tag{21}
$$

Note that the delta distribution does not appear in the commutator susceptibility as it cancels $\chi^c(\omega) = \frac{1}{i}(\chi^{>}(\omega) - \chi^{<}(\omega))$. This is not the case for the anti-commutator (or Keldysh component) of the Green's function:

$$
\Psi_{AB}(\omega) = \chi^{>}(\omega) + \chi^{<}(\omega) = 4\pi i C \delta(\omega) + i\mathcal{P}\chi^c(\omega)\coth(\beta\omega/2).
\tag{22}
$$

Equation (22) can be regarded as an extension of the fluctuation-dissipation theorem [26] to a specific case of nonergodic systems in the sense of Eq. (6) (see also [24]). If present $C \neq 0$ the singular term should be treated explicitly. It has been noted in a recent analysis performed on the real frequency axis by the Schwinger-Keldysh formalism that incorporating this term into the retarded or advanced component leads to wrong results even for the simplest case of the atomic limit [27, App. E].

### 2.2.2 Thermal Green's function

The thermal- or Matsubara- Green's function for the (bosonic) observable $\hat{A}, \hat{B}$ is defined as

$$
\chi^{\text{th}}(\tau) = \langle \mathcal{T}\hat{A}(\tau)\hat{B}(0)\rangle,
\tag{23}
$$

where $\tau$ denotes the Wick-rotated imaginary time ($t \rightarrow -i\tau$ such that $\hat{A}(\tau) = e^{\tau\hat{H}}\hat{A}e^{-\tau\hat{H}}$), and $\mathcal{T}$ is the corresponding (imaginary) time-ordering operator. The thermal Green's function is defined for $\tau \in [-\beta, \beta]$ and fulfills

$$
\chi^{\text{th}}(\tau < 0) = \chi^{\text{th}}(\tau + \beta),
\tag{24}
$$

which corresponds to Eq. (13). It can be expanded in a Fourier-series

$$
\begin{aligned}
\chi^{\text{th}}(\tau) &= \frac{1}{\beta}\sum_{n=-\infty}^{\infty}\chi^{\text{th}}(i\omega_n)e^{-i\tau\omega_n}, \\
\chi^{\text{th}}(i\omega_n) &= \int_0^{\beta}d\tau\,\chi^{\text{th}}(\tau)e^{i\tau\omega_n} \\
&= \int_0^{\beta}d\tau\,\langle\hat{A}(\tau)\hat{B}(0)\rangle e^{i\tau\omega_n},
\end{aligned}
\tag{25}
$$

where $\omega_n = 2\pi T n$ are the (bosonic) Matsubara frequencies (with $n \in \mathbb{Z}$). For deriving an integral representation of $\chi^{\text{th}}(i\omega_n)$, it is useful to reverse the Wick-rotation by the variable substitution $\tau = it$. By exploiting the analytic properties of $\chi^{>}(t)$, inserting Eq. (21) and performing the integration over the complex time variable yields the desired integral representation [3]:

$$
\chi^{\text{th}}(i\omega_n) = C\beta\delta_{n,0} - \frac{1}{2\pi}\mathcal{P}\int_{-\infty}^{\infty}\frac{\chi^c(\omega)}{i\omega_n - \omega}d\omega.
\tag{26}
$$

The Matsubara or thermal susceptibility contains information on *both*, the static isothermal susceptibility $\chi^T = \chi^{\text{th}}(i\omega_n = 0)$ (cf. [19, Sec. 2]) and the Kubo dynamical susceptibility $\chi^{\mathcal{R}}(\omega)$.

---

[3]There is an error in [20, Eq. 4.4-4.5]. The correct result is given in Eq. (26)

It should be noted that the regular part (second term in Eq. (26)) can be analytically continued to the entire complex plane, except for the real axis where it has a branch cut. Hence, in general it is not correct to simply replace $i\omega_n$ with $\omega+i0^+$ for the full thermal Green's function $\chi^{\text{th}}(i\omega_n)$ (see also [20, 28]). The correct procedure for the analytic continuation requires to first remove the anomalous part, and then make the replacement:

$$\chi^{\mathcal{R}/\mathcal{A}}(\omega) = \left(\chi^{\text{th}}(i\omega_n) - \beta C \delta_{n,0}\right)_{i\omega_n \to \omega \pm i0^+} \tag{27}$$

or equivalently

$$\chi^c(\omega) = \left(\chi^{\text{th}}(i\omega_n) - \beta C \delta_{n,0}\right)_{i\omega_n \to \omega + i0^+} - \left(\chi^{\text{th}}(i\omega_n) - \beta C \delta_{n,0}\right)_{i\omega_n \to \omega + i0^-}. \tag{28}$$

For the case considered in this work, $\hat{A} = \hat{B}$ and with time-inversion symmetry, the Cauchy principal value in Eq. (26) is not necessary. From the definitions in Eq. (10) it is also easy to see that for this case $\chi^c(t) \in i\mathbb{R}$ and $\chi^c(-t) = -\chi^c(t)$ and therefore $\chi^c(\omega) \in \mathbb{R}$ and $\chi^c(-\omega) = -\chi^c(\omega)$. By symmetry it follows that $\chi^c(\omega) = 2\Im\chi^{\mathcal{R}}(\omega)$. Thus Eq. (26) simplifies to

$$\chi^{\text{th}}(i\omega_n) = C\beta\delta_{n,0} + \frac{2}{\pi}\int_0^\infty \frac{\omega}{\omega_n^2 + \omega^2}\Im\chi^{\mathcal{R}}(\omega)\,d\omega. \tag{29}$$

This is the expression we use later on for the spin-spin susceptibility in the Hubbard model.

### 2.2.3 Lehmann representation of $C$

The thermal Green's function may also be written as

$$\chi^{\text{th}}(i\omega_n) \quad = \chi^{\text{th}}_{\text{reg}}(i\omega_n) + \delta_{n,0}\beta C, \tag{30}$$

where the explicit expression of the singular (Kronecker-delta) term reads

$$C = \frac{1}{Z}\sum_{\substack{l,m \\ E_l = E_m}} e^{-\beta E_l} A_{lm} B_{ml}, \tag{31}$$

where the Lehmann summation includes only the terms with $E_l = E_m$, encoding, beyond the diagonal contributions, all the possible *degeneracies* in the energy spectrum. As mentioned before, we assume here and thereafter that $\langle\hat{B}\rangle = 0$. If this is not the case an additional term $\langle\hat{A}\rangle\langle\hat{B}\rangle$ needs to be subtracted from Eq. (31). While being consistent with the spirit of the LRT, adopting this definition also allows to identify insightful links between the coefficent $C$ and the intrinsic properties of the underlying many-particle energy spectrum. For instance, it can be readily seen that at zero temperature *only* a degenerate groundstate can lead to $C \neq 0$, as any nondegenerate groundstate results in $C = A_{00}B_{00} - \langle A\rangle\langle B\rangle = A_{00}B_{00} - A_{00}B_{00} = 0$. At finite temperatures, instead, a rigorous relation between a non-vanishing $C$ and the presence of degeneracies cannot be established in general, because diagonal elements ($l = m$) of the excited part of the many-electron spectrum might also contribute to a $C \neq 0$. However, if either $\langle\hat{A}\rangle$ or $\langle\hat{B}\rangle$ is *independent* of temperature prior to its redefinition ($\hat{A} \to \hat{A} - \langle\hat{A}\rangle$)[4] then the purely diagonal terms $l = m$ in Eq. (31) *cannot* yield any contribution to $C$. Indeed, in such cases the condition $\langle\hat{A}\rangle = \sum_l \frac{e^{-\beta E_l}}{Z} A_{ll} = 0$ for all temperatures implies that for each energy-subspace one must have $\sum_{l'} A_{l'l'} = 0$. Hence, in the non-degenerate case, when the energy-subspace is spanned by a single eigenstate, one will always find $A_{ll} = 0$, with the following relevant implication: If $C \neq 0$ the many-electron Hamiltonian must have at least one *degenerate* energy

---

[4]Or, equivalently the matrix elements of $\hat{A}$ (or $\hat{B}$) do not depend on temperature after the subtraction of $\langle\hat{A}\rangle$ (or $\langle\hat{B}\rangle$).

level. It is important to emphasize, here, that the condition of a temperature independence of $\langle A \rangle$ (or of $\langle B \rangle$), under which a direct link between a non-vanishing $C$ and the degeneracies of the many-electron Hamiltonian holds at *all* temperatures, being tied to the symmetries of the problem under investigation, is verified in several situations relevant for the condensed matter theory, such as, e.g., for both the uniform and the local magnetic or density responses of non long-range ordered many-electron systems. Evidently, this also applies to the specific case considered in our work, where $\hat{A} = \hat{B}$ corresponds to the local magnetic moment operator $\hat{M}_z = g\hat{S}_z = n_\uparrow - n_\downarrow$ (with $g = 2$) of the Hubbard model, for which $\langle \hat{S}_z \rangle = 0$ at all temperatures, due to the SU(2)-symmetry of the problem.

The regular part of Eq. (30) is given by

$$\chi^{\text{th}}_{\text{reg}}(i\omega_n) \quad = {\sum_{lm}}' \, e^{-\beta E_l} A_{lm} B_{ml} \frac{e^{\beta E_{lm}} - 1}{i\omega_n + E_{lm}}, \tag{32}$$

where the symbol $\sum'$ denotes the (corresponding) exclusion of all singular terms with $E_l = E_m$ at zeroth Matsubara frequency.

The link between the value of $C$ and the possible presence of degeneracies of the many-body Hamiltonian reflects, more in general, the role played by the symmetries in triggering non-ergodic behaviors [24]. Specifically, if $\hat{\Omega}_j$ are all constants of motion of a system ($\frac{d}{dt}\hat{\Omega}_j(t) = i[\hat{H}, \hat{\Omega}_j] = 0$ for all $j$) then

$$C = \lim_{\mathcal{T} \to \infty} \frac{1}{\mathcal{T}} \int_0^{\mathcal{T}} dt \, \langle \hat{A}\hat{B}(t) \rangle = \sum_{j=1}^{\infty} \frac{\langle \hat{A}\hat{\Omega}_j \rangle \langle \hat{B}\hat{\Omega}_j \rangle}{\langle \hat{\Omega}_j^2 \rangle} \, . \tag{33}$$

One operator that always commutes with the Hamiltonian is $\Delta\hat{H} = \hat{H} - \langle \hat{H} \rangle$. This "trivial" symmetry is conventionally associated with the first term ($j = 1$) in the sum of the r.h.s. of Eq. (33). In fact, one can show [19, 24] that this term is precisely the one responsible for the difference between the isentropic and the isothermal susceptibility

$$\chi^T - \chi^S = \beta \frac{\langle \hat{A}\Delta\hat{H} \rangle \langle \hat{B}\Delta\hat{H} \rangle}{\langle (\Delta\hat{H})^2 \rangle}, \tag{34}$$

generally described by Eq. (3).

This aspect has important implications for the case mostly considered in our work ($\hat{A} = \hat{B} = \hat{S}_z$): Since $\langle \hat{n}_\uparrow \Delta\hat{H} \rangle = \langle \hat{n}_\downarrow \Delta\hat{H} \rangle$, we obtain that $\chi^T = \chi^S$, i.e., the measurement of the local spin response of the Hubbard model does *not* distinguish between adiabatic and thermal processes. The same conclusion does not necessarily apply, as we will see below, to the case of the Kubo (i.e., isolated) measurements.

### 2.2.4 Atomic limit

Considering the limiting case of a single site with local repulsion ($H_{\text{at}} = U\hat{n}_\uparrow\hat{n}_\downarrow - \mu(\hat{n}_\uparrow + \hat{n}_\uparrow)$) provides a first, simple but instructive example for applying the formalism introduced above and for understanding its physical implications. (Where the chemical potential $\mu = U/2$ at half-filling.) We consider the operators $\hat{A} = \hat{B} = g\hat{S}_z$ (where $g = 2$). Indeed, since $[\hat{H}_{\text{at}}, \hat{S}_z] = 0$, all the susceptibilities involving $\hat{S}_z$ only are purely static. The corresponding commutator susceptibility ($\chi^c$) is thus *identically* zero. Hence, it follows that the advanced and retarded Kubo susceptibility as well as the regular part of the thermal Green's function $\chi^{\mathcal{R}}(t) = \chi^{\mathcal{A}}(t) = \chi^c(t) = \chi^{\text{th}}_{\text{reg}}(i\omega_n) \equiv 0$. From the imaginary time perspective this reflects into the following behavior:

$$\chi^{\text{th}}(\tau) = g^2 \langle \hat{S}_z^2 \rangle \iff \chi^{\text{th}}(i\omega_n) = \beta g^2 \langle \hat{S}_z^2 \rangle \delta_{n,0} \, .$$

Then, the only not-vanishing contribution on the Matsubara frequency axis is the singular part $C = g^2 \langle \hat{S}_z^2 \rangle$. Physically, this encodes the physics of a "perfect" isolated magnetic moment, whose isothermal susceptibility, given by the zeroth Fourier component ($\omega_0 = 0$) of the thermal Green's function, shows the corresponding Curie behavior $\chi^T = \frac{g^2 \langle \hat{S}_z^2 \rangle}{T}$. The Keldysh or anti-commutator component of the Green's function reads in time as $\Psi(t) = \text{const.} = 2g^2 i \langle \hat{S}_z^2 \rangle$ and in frequency $\Psi(\omega) = 4\pi i C \delta(\omega) = 4\pi i g^2 \langle \hat{S}_z^2 \rangle \delta(\omega)$. Equation (33) demonstrates that this behavior is linked, logically, to SU(2)-symmetry of the problem, and precisely to the constant of motion $S_z$ itself.

# 3 Model and Methods

## 3.1 Model

For our numerical study, whose results are shown in Section 4, we consider the half-filled Hubbard model on a Bethe lattice. This can be solved exactly at finite temperature $T$ by means of Dynamical Mean Field Theory (DMFT) [2], which features a prototypical description [29,30] of the nonperturbative [31–34] physics of the Mott-Hubbard MIT. Here, we only consider paramagnetic DMFT solutions, disregarding the onset of antiferromagnetic order at low-temperatures.[5]

The Hamiltonian reads:

$$\hat{H} = -\sum_{\langle ij \rangle, \sigma} t_{ij} \left( \hat{c}_{i\sigma}^\dagger \hat{c}_{j\sigma} + \hat{c}_{j\sigma}^\dagger \hat{c}_{i\sigma} \right) + U \sum_i \hat{n}_{i\uparrow} \hat{n}_{i\downarrow}, \tag{35}$$

where $t_{ij} = \frac{t}{z}$ is the nearest-neighboring hopping amplitude from site $i$ to $j$ of the lattice, $z$ is the coordination number and $U$ is the local interaction. In the limit of large $z$ (where DMFT becomes exact) the corresponding density of states is semi-circular

$$\mathcal{N}(\epsilon) = \frac{1}{2\pi t^2} \sqrt{4t^2 - \epsilon^2}, \quad |\epsilon| < 2t. \tag{36}$$

Throughout the paper, we fix $t = 0.5$, so that the half-bandwidth ($W/2 = 1$) of the DOS sets our energy units.

## 3.2 DMFT

The DMFT simulation was performed with a continuous-time quantum Monte Carlo (QMC) algorithm implemented in the code package `w2dynamics` [35]. For converging the one-particle properties symmetric improved estimators [36] where used. After converging the DMFT calculations on the one particle-level for each parameter set, we have measured the (thermal) spin-spin susceptibility in imaginary time in the segment implementation [35]. More details on the specific parameters and settings are reported in Appendix B. All real frequency data was obtained by analytic continuation. The input to the analytic continuation procedure, $\chi_{S_z S_z}^{\text{th}}(i\omega_n)$, came from the QMC-solver which gives intrinsically noisy results. In the following we describe the details of the analytic continuation.

---

[5]We note that this assumption, which is often made within DMFT studies of the Mott-Hubbard MIT, would rigorously hold for a Bethe lattice with random hopping, which has the same density of states and thus DMFT solution as the regular Bethe lattice in the paramagnetic phase [2].

### 3.3 Numerical analytic continuation of $\chi_{S_z S_z}(\mathrm{i}\omega_n)$

For the analytic continuation of a (possibly) non-ergodic system (i.e., with $C > 0$), the main task is evidently to separate the anomalous part ($\propto C$) from the *regular* part $\chi_{\mathrm{reg}}^{\mathrm{th}}(\mathrm{i}\omega_n)$. Formally this goal can be achieved by computing

$$C = \frac{1}{\beta}\left(\chi^{\mathrm{th}}(\mathrm{i}\omega_n = 0) - \Re\chi^{\mathcal{R}}(\omega = 0)\right). \tag{37}$$

At $\omega = \mathrm{i}\omega_n = 0$ the integral representation of the regular part of the thermal Green's function is equivalent to the Kramers-Kronig relation

$$\Re\chi^{\mathcal{R}}(\omega = 0) = \chi_{\mathrm{reg}}^{\mathrm{th}}(\mathrm{i}\omega_n = 0) = \frac{1}{\pi}\int_{-\infty}^{\infty}\mathrm{d}\omega'\frac{\Im\chi^{\mathcal{R}}(\omega')}{\omega'}. \tag{38}$$

Analytically, this amounts to replace $\mathrm{i}\omega_n$ with $\omega + \mathrm{i}0^+$ in $\chi^{\mathrm{th}}(\mathrm{i}\omega_n)$ for the *positive* Matsubara frequencies only.

Numerically, this would correspond to extrapolating the Matsubara frequency data from $\omega_n > 0$ to $\omega_n = 0$ [6].

$$\chi^{\mathrm{th}}(\mathrm{i}\omega_n > 0) = \frac{2}{\pi}\int_0^{\infty}\mathrm{d}\omega\frac{\omega}{\omega^2 + \omega_n^2}\Im\chi^{\mathcal{R}}(\omega). \tag{39}$$

While it is well-known that the problem of analytic continuation of numerical data from the Matsubara to the real frequency axis is formally ill-defined, highly precise calculations at very low Matsubara frequencies might allow to invert Eq. (39) at an acceptable degree of precision in physically relevant real-frequency intervals, by exploiting the Padé interpolation procedure [38] or more advanced, recently introduced [39] approaches. In those cases, if the results of, e.g. the Padé approximation, significantly change depending on whether the zeroth Matsubara frequency is excluded or included, the presence of an anomalous (or non-ergodic) contribution $C > 0$ in the data should be supposed. Its magnitude can be then directly quantitatively estimated by evaluating Eq. (37) through the corresponding Padé approximants.

For less precise or intrinsically noisy numerical data other methods for the analytic continuation are needed. Indeed, a variety of methods [40–43] are available. In principle all of them work in both cases, i.e., including or excluding the zeroth Matsubara frequency, which is an essential prerequisite for our study. In practice, as our DMFT calculations exploit a QMC impurity solver, we have mostly used the Maximum Entropy (MaxEnt) method.

While the general MaxEnt procedure is broadly known [40], it is worth to discuss here its specific adaption to the precise scopes of our analysis. Formally, we try to solve a Fredholm integral of the first kind

$$\chi^{\mathrm{th}}(\mathrm{i}\omega_n) = \int_0^{\infty}\mathrm{d}\omega\,K(\omega_n,\omega)A(\omega)\quad n > 0\,, \tag{40}$$

where the l.h.s. of the equation is the (statistically noisy) input of our QMC solver and the kernel $K(\omega_n,\omega)$ is precisely know a priori, to determine the spectral function $A(\omega)$. Note that, for a better numerical treatment, a factor of $\omega^{-1}$ was absorbed into $A$ to make both quantities finite for all $\omega, \omega_n$:

$$A(\omega) := \frac{2}{\pi}\frac{\Im\chi^{\mathcal{R}}(\omega)}{\omega}\quad\text{and}\quad K(\omega_n,\omega) := \frac{\omega^2}{\omega_n^2 + \omega^2}. \tag{41}$$

---

[6] In the Kondo-Bose model also a finite $C$ was found [37, Fig. 3] which was extracted by a quadratic fit through the first three positive Matsubara frequencies.

In this respect, the norm of $A$ is of interest as $C = \frac{1}{\beta}\left(\chi^{\text{th}}(\omega_n = 0) - \int_0^\infty A(\omega)\mathrm{d}\omega\right)$. The main challenge to the applications of this procedure is, however, that the rigorous exclusion of the purely static term $\chi^{\text{th}}(0)$ from Eq. (40) intrinsically limits the definition of the spectrum $A(\omega)$ in the lowest-frequency regime: $A(\omega)$ is defined only up to peak-structures located at $|\omega| \ll 2\pi T$ with a width $\ll 2\pi T$, as their presence would only affect the zeroth Matsubara frequency term. At sufficiently low $T$, this problem of the MaxEnt scheme can be remedied by only allowing for peaks with a width larger than $2\pi T$. In the context of image reconstruction, this procedure was introduced by Skilling [44, Ch. 2.4] and Gull [45, p. 53-73]. Specifically, one requires the spectral function $A(\omega)$ to be a convolution of a hidden spectral function $h(\omega)$ with a Gaussian broadening $g_b(\omega)$:

$$A(\omega) = g_b(\omega) * h(\omega) \quad \text{with} \quad g_b(\omega) = \frac{1}{\sqrt{2\pi}b}\exp[-\omega^2/(2b^2)]\,, \tag{42}$$

which described the effective "blurring" of our spectral function associated with the finite precision of our input data. In our case, the corresponding blur width $b$ must be required to be larger than $\pi T$. This corresponds, in practice, to evaluate the entropy term of MaxEnt on the hidden spectrum $h$. Following this approach, known as *preblur* scheme, one minimizes $Q[h]$ with respect to $h$, where

$$
\begin{aligned}
Q[h] &= \tfrac{1}{2}\chi_{\text{reg}}^2[g_b * h] + \alpha S[h]\,,\\
\chi_{\text{reg}}^2[A] &= \sum_{n>0}\left|\frac{\chi^{\text{th}}(i\omega_n) - \int_0^\infty \mathrm{d}\omega\, K(\omega_n,\omega)A(\omega)}{\sigma(\omega_n)}\right|^2\,,\\
S[h] &= \int_0^\infty\left[h(\omega) - D(\omega) - h(\omega)\log\frac{h(\omega)}{D(\omega)}\right]\,.
\end{aligned}
\tag{43}
$$

For computing the results shown in the next section, we used the open-source implementation of MaxEnt (including the preblur) of [41]. In particular, the hyper-parameter $\alpha$ has been determined though the `chi2kink` method of [41] and references therein. In Eq. (43) $\sigma(\omega_n)$ is the QMC estimate of the error [7]. As a default model $D(\omega)$ for the MaxEnt a constant has been assumed, after verifying that the final results do not appreciably change by using Lorenzian and Gaussian default models.

*Extraction of the maximally allowed peak width* – In order to distinguish the regions of the phase diagram where $C \neq 0$ from those where $C = 0$, which is one of the main goals of our numerical study, it is crucial to verify whether the (full) QMC data set, *including* the zeroth Matsubara frequency, can also be explained by regular contributions *only*. In practice, one has to carefully check the effect of a finite blur width on the fitting procedure: If the blur width has little influence on the quality of the fit for $b < a\pi T$ with e.g. $a = 0.5$, one should conclude that the obtained QMC data can also be explained in terms of a regular contribution only (i.e., $C = 0$). On a more quantitative perspective, by finding the point at which the smallest obtainable value of $\chi^2$ rises steeply, one can set a lower bound for the life-time of the excitations under consideration (spin-excitations in our case).

*Note on general case* – While we will focus in the following on the case $\hat{A} = \hat{B} \propto S_z$, the procedure described here could be also applied to the generic case of two different observables $\hat{A}$ and $\hat{B}$. In such a case, one should consider that the spectral function in Eq. (26) contains a real as well as an imaginary part. Using the Kramers-Kronig relations one can show that the imaginary part gives the same contribution as the real part. The imaginary part of the $\chi^{\mathcal{R}}(\omega)/\omega$ is, in contrast to the case where $\hat{A} = \hat{B}$, not positive semi-definite. MaxEnt as well

---

[7]In general the covariance matrix can be obtained by a bootstrap algorithm (see for instance [46]). For our case this was however unfeasible. We used a constant for $\sigma(i\omega_n)$ instead. This corresponds to uncorrelated QMC noise in imaginary time (by Gaussian error propagation). Note also that using `chi2kink` for the hyper-parameter determination makes the MaxEnt result invariant under rescaling the error.

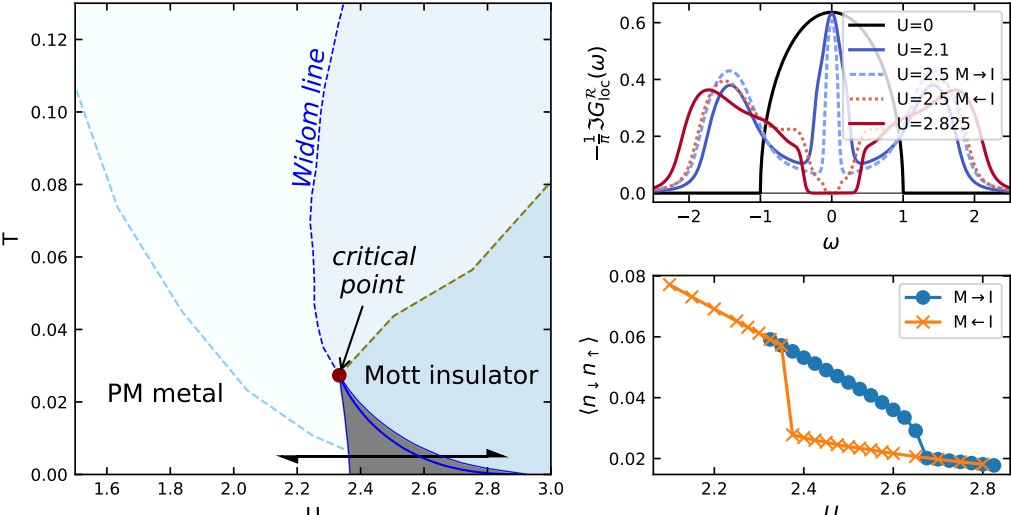

Figure 1: *Left:* Schematic DMFT phase diagram of the half-filled Hubbard model on a Bethe lattice with unitary half bandwidth. The thick solid blue line marks the first order MIT, $U_c(T)$, the gray-shadowed the coexistence region between $U_{c1}(T)$ and $U_{c2}(T)$, both taken from [29]. The local moment formation, as estimated in [15][9], is marked by the azure dashed line; the Widom line (blue dashed) is taken from [48], while the opening of the Mott gap (olive dashed) has been estimated by [49]. The black double arrow marks the parameter paths we focus on in our work. *Right*: Evolution of the local spectral function (upper panel) and of the double occupancy (lower panel) computed in DMFT along the path shown in the phase diagram on the left.

as the *preblur* method can however also be used for the general case. Skilling [44, Ch. 2.5] argued that the hidden image $h$ should be related to $A$ by an orthogonal transformation. One can then split the hidden image into two positive semi-definite parts $h^+(\omega)$ and $h^-(\omega)$ each with an entropy term given by Eq. (43). The orthogonal transformation $A(\omega) = h^+(\omega) - h^-(\omega)$ is contained as a special case in class of transformations Skilling proposed. MaxEnt is therefore also applicable to a general susceptibility. The problem of a possible jump in the zeroth Matsubara frequency needs to be addressed the same way we do in this paper [8].

## 4 Numerical results within the coexistence region

In order to place our analysis of the long-term memory spin-correlations in the proper physical context, we concisely summarize in Fig. 1 the Mott-Hubbard metal-insulator (MIT) transition, as described by DMFT. In the main panel of the figure, we sketch the paramagnetic phase diagram of the Hubbard model on a Bethe-lattice, highlighting the essential features known to characterize the Mott MIT: The first order transition is marked by a blue solid line ending at the critical point (red dot) and the gray-shadow area corresponds to the coexistence region, where two independent DMFT solutions associated to the hysteresis of the transition are found. The data for the first order phase transition were determined [29] through the criterion of

---

[8]For very precise data one could also utilize an interpolation algorithm based on the Carathéodory functions [47]. Also there one has to be aware of the effect of the possible jump of the zeroth Matsubara frequency.

[9]Converted from 3D lattice to Bethe lattice by resealing with $\frac{1}{2\sqrt{6}}$ to obtain a density of states with the same variance.

the equality of the free energies in the insulating and in the metallic solution. Consistent with the recent literature [15, 48, 49], the smooth high-$T$ crossover from a good metallic to a bad metallic, and eventually to an insulating behavior has been characterized through the progressive fulfillment of specific conditions (dashed lines), describing (i) the formation of the local moments (light blue) [15], (ii) the onset of well-defined Mott properties (olive) [49], and (iii) the so-called Widom condition (blue) [48], respectively. The latter is defined by the parameters above the critical point corresponding to a relative minimum of the thermodynamic stability of the system, i.e., more precisely, via the condition to be equally close to both the metallic and the insulating phase. In practice, the fulfilment of the Widom condition can be determined from the curvature of the free-energy functional at its minimum [48] and, hence, via the analysis of the Jacobian of the DMFT fix point function [50]. Physically, the Widom line represents a natural prolongation of the true MIT line at higher temperatures. Other criteria based on the double-occupancy are also possible and lead to similar results [49].

On a more quantitative level, we illustrate the well-know [2] evolution of the Mott MIT physics occurring along the parameter paths marked by the double-headed black arrow (fixed low $T$: $\beta = 200$, and $2.1 < U < 3.8$) by hands of our DMFT calculations of the local spectral function (upper right panel) and of the double occupancy (lower right panel). The considerably different evolution of such quantities along the chosen paths highlights the 1st order nature of the MIT: In the whole coexistence region along the path of slowly increasing $U$ values ($M \rightarrow I$) both the spectral function at the Fermi level and the double occupancy yield significantly larger (i.e., "metallic"-like) values, compared to the results obtained for the same parameter along the reversed path ($I \rightarrow M$).

Eventually, before presenting our numerical results, we want to emphasize that DMFT for the Hubbard model corresponds to its exact solution in the limit of infinite dimensions (or lattice coordination number). Certain aspects of the underlying physics are expected to qualitatively change in finite dimensional cases. This certainly applies to the nature of the Mott insulating ground state, which is highly degenerate in paramagnetic DMFT calculations. Nonetheless, the clear-cut nature of the DMFT description in infinite dimensions makes it an ideal test bed for illustrating the applications of our algorithmic procedure and the physical interpretation of obtained results, paving the way for future studies including the effects of non-local correlations e.g., via cluster [51] and diagrammatic [52] extensions of DMFT and/or the onset of long-range ordering at low-$T$ [53, 54].

## 4.1 Calculations on imaginary time axis

Starting from our converged DMFT solution, we calculated the local spin-susceptibility in the imaginary time domain $\chi^{\mathrm{th}}(\tau) = \langle \mathcal{T}\hat{M}^z(\tau)\hat{M}^z(0)\rangle$ with $\hat{M}^z = g\hat{S}_z$ and $S_z = \frac{1}{2}(\hat{n}_\uparrow - \hat{n}_\downarrow)$ (with $\hbar = 1$ and the electronic gyromagnetic factor set to $g = 2$) as well as its Fourier transform in Matsubara and real frequencies. We begin by discussing our results in (imaginary) time domain, reported on the left panels of Fig. 2, as they represent the typical output of a DMFT(QMC) calculation. We recall that, as the two observables in our correlation function coincide ($\hat{A} = \hat{B} = \hat{M}_z$), $\chi^{\mathrm{th}}(\tau)$ will be symmetric with respect to $\beta/2$ in the interval $\tau \in [0, \beta)$, s. e.g. [13, Ch. 2.3]- [54, App. B], allowing us to restrict the analysis to the interval $[0, \beta/2]$. The upper (bottom) panel of the left part of Fig. 2 shows the evolution of $\chi^{\mathrm{th}}(\tau)$ along the $M \rightarrow I$ ($M \leftarrow I$) parameter path cutting the coexistence region at a fixed $T$ ($\beta = 200$) for increasing (decreasing) interaction values, highlighted by the black arrow in Fig. 1. The underlying first order MIT, as well as its associated hysteresis, directly reflect into the abrupt changes displayed by $\chi^{\mathrm{th}}(\tau)$. Specifically, even at first glance, the results display two *qualitatively different* behaviors: (i) for the $U$ values highlighted by red-colours, which correspond in both panels to the insulating solutions, $\chi^{\mathrm{th}}(\tau)$ is characterized by a rapid decay [2, 5] to a rather large constant offset, while (ii) for lower $U$ (bluish colors) a significantly slower decay [2, 5]

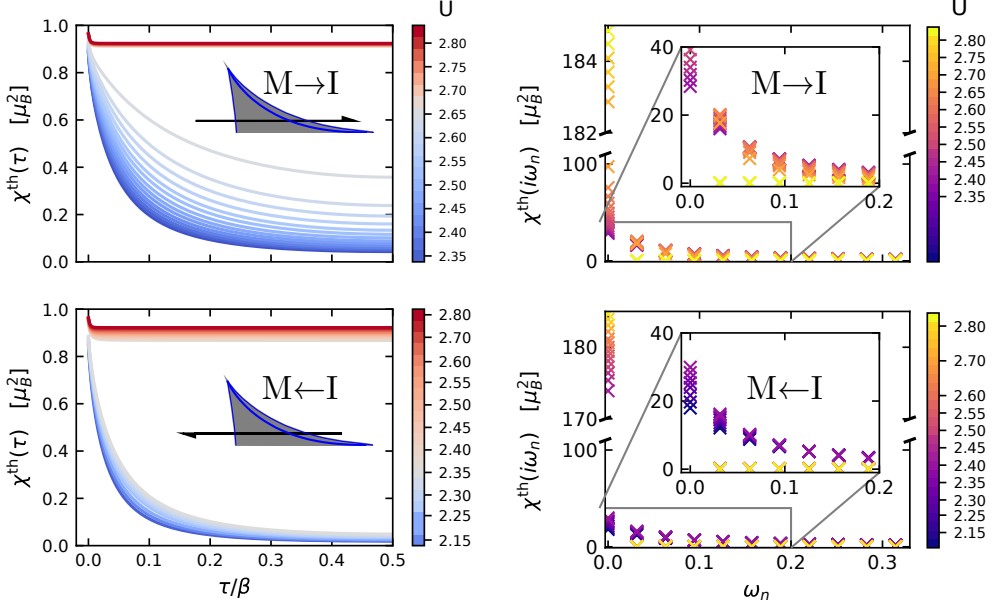

Figure 2: Spin susceptibility computed in DMFT as a function of imaginary time (left) and Matsubara frequencies (right) for $T = \frac{1}{200}$ through the coexistence region of the Mott MIT in the half-filled Hubbard model. For $\chi^{\text{th}}(\tau)$ the colorscale encoding the different values of $U$ was set such that the grey color marks the crossing of the corresponding transition lines, i.e. $U_{c2}(T=1/200)$ for the $M \to I$ path (top panel) and $U_{c1}(T=1/200)$ for the $M \leftarrow I$ path (bottom panel). The presence of an anomalous term $C \neq 0$ should be suspected if a constant contribution to $\chi^{\text{th}}(\tau)$ (left) or a discontinuity of the value at the zeroth Matsubara frequency (right) is identifiable in the data.

towards much smaller values of the dynamical susceptibility at $\tau = \frac{\beta}{2}$ is observed. The values of $\chi^{\text{th}}(\tau = \frac{\beta}{2})$ display, nonetheless, a sizable increase for the highest $U \lesssim U_{c2}$ values along the first path ($M \to I$), where a (metastable) metallic solution can be still obtained (bluish-grey colors). Evidently, the two classes of well distinct susceptibility behaviors directly encode the different dynamic properties of local magnetic moments in the Mott insulating and correlated metallic phases and, in particular, the completely different impact of screening processes in these two cases.

In order to go beyond these qualitative considerations, one needs to quantify the size of possibly emergent long-term memory (or non-ergodic) effects in the parameter region of the phase diagram, where the screening mechanisms are working poorly. In general, this piece of information cannot be extracted directly[10] from the corresponding value of $\chi^{\text{th}}(\tau = \frac{\beta}{2})$, which is always finite at finite $T$ even for $U = 0$. The value at $\tau = \beta/2$ is in general given by

$$\chi^{\text{th}}(\tau = \beta/2) = C + \frac{1}{4\pi}\mathcal{P}\int_{-\infty}^{\infty} d\omega\, \chi^{c}(\omega)\,\text{sech}(\beta\omega/2), \qquad (44)$$

i.e. it also has a contribution (for finite temperature) if $C = 0$.

In this perspective, it may be more convenient to extract the anomalous part ($C$) from the Matsubara frequency behavior of $\chi^{\text{th}}$. In the Matsubara representation, the differences between normal part and anomalous part of the dynamic susceptibility can be directly visualized by including/excluding $\chi^{\text{th}}(i\omega_n = 0)$ from the data analysis.

---

[10]For a detailed discussion of the regime of applicability of this procedure s. Ref. [5]

## 4.2  Data on Matsubara frequency axis and analytic continuation

In the right part of Fig. 2 we show our DMFT results for the thermal spin-susceptibility Fourier-transformed to Matsubara frequencies. For low temperatures, as $T = 1/200$ considered here, Fig. 2, it is possible to estimate the magnitude of $C$ from the discontinuity of $\chi^{\text{th}}(i\omega_n)$ at $i\omega_n = 0$ quite reliably even by the naked eye. However, for higher temperatures and/or more border-line cases (e.g., where $C$ is small), more refined treatments are needed. In such cases, it would be first necessary to determine in the most rigorous way, whether an anomalous contribution $C \neq 0$ is present in the data. To this aim, we will introduce a specific procedure, based on a detailed inspection of the evolution of the minimal value of the fit-loss functional

$$\chi^2[A, b] = \sum_{n \geq 0} \left| \frac{\chi^{\text{th}}(i\omega_n) - \int_0^\infty d\omega\, K(\omega_n, \omega) \left[ g_b * A \right](\omega)}{\sigma(\omega_n)} \right|^2 , \tag{45}$$

as the blur width $b$ is varied (minimized with respect to $A(\omega)$). In contrast to Eq. (43) the zeroth Matsubara frequency is now included. The functions ($K$, $g_b$ and $\sigma$) are the same as in Section 3.3.

The idea behind this approach is the following: Mathematically, any observed difference between the value of $\chi^{\text{th}}$ at the zeroth ($i\omega_0 = 0$) and the first ($i\omega_1 = i\pi T$) Matsubara frequency can always be formally explained either by a true anomalous term ($C > 0$) or by a sufficiently narrow ($\ll \pi T$) low-energy peak in $A(\omega) \propto \Im \chi^{\mathcal{R}}(\omega)$. However, if the former feature ($C > 0$) is actually present in the raw data, by tuning the blur parameter $b$ over the scale $\pi T$, the nonzero Matsubara frequencies should also become progressively affected by its presence. As no anomalous term is included in the regular part of the spectrum, this would then prevent the possibility of performing a good fit, corresponding to a *steep increase* of min $\chi^2[A, b]$ for $b \gtrsim \pi T$. On the contrary, if $C = 0$, $A(\omega)$ already includes the whole spectral weight small modulations of $b$ will *not* have significant effects on the fit accuracy, featuring a rather smooth behavior of min $\chi^2[A, b]$.

Hence, by inspecting the behavior of min $\chi^2[A, b]$ as a function of $b$, one can clearly differentiate between contributions to $\chi^{\text{th}}(i\omega_n = 0)$ arising from the anomalous term ($C > 0$) and those purely explained by low-energy features of the regular spectral function.

In Fig. 3 we show the application of this procedure to one of the parameter paths (i.e., $M \to I$) in the coexistence region discussed above. In particular, we report there the variation of the minimum of fit loss expression as a function of blur width $b$, for the temperature $T = 1/200$ and four different $U$-values, of which two have been chosen very close to the corresponding $U_{c2}(T = 1/200) = 2.66$ threshold. We observe that only for the insulating solutions, i.e. those found for $U \geq 2.675$, min $\chi^2[A, b]$ becomes strong $b$-dependent, indicating the impossibility of a reliable fit when $b$ is increased. On the contrary, for the other two values of $U \leq U_{c2}(T = 1/200)$ the behavior min $\chi^2[A, b]$ is essentially unaffected by the modulation of $b$ over a quite large scale. On the basis of these results, we conclude that, our low-$T$ data are consistent with the presence of an anomalous term ($C \neq 0$) exclusively in the Mott insulating solutions.

## 4.3  Spectral functions and spectral weights

By exploiting the information gained about the anomalous term, we can now proceed to perform the analytic continuation onto the real-frequency axis in the most rigorous way. In particular, by analytically continuing our data along the path at $T = 1/200$, we have set $C \equiv 0$ for all thermodynamically stable metallic solutions (i.e. for all $U < U_c(T = 1/200)$ shown in the left panel of Fig. 4), which corresponds to include the zeroth Matsubara frequency in the data set. Obviously, the same assumption does not hold for the insulating cases for $U > U_c(T = 1/200)$,

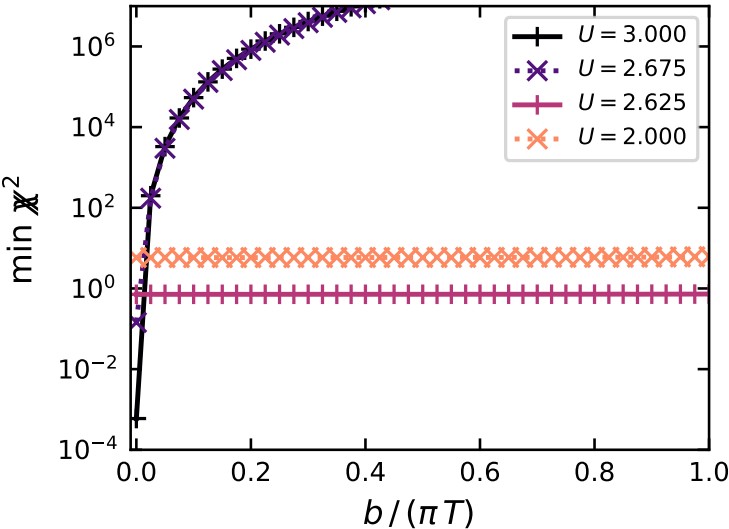

Figure 3: Minimum values of the fit loss as a function of the blur parameter $b$ for different interaction values and $T = 1/200$. A strong(very weak) dependence of this quantity as a function of $b$ is a reliably rigorous marker for the presence(absence) of an anomalous term in the response function.

right panel. The spin absorption spectra reported in Fig. 4 evidence the progressive softening (as well as the simultaneous narrowing) of the peak by increasing $U$ in the metallic phase, which hints at the progressive slowing-down of the local spin fluctuations expected [5,7] by approaching the MIT at low-$T$. However as $T$, though small, is finite, the Mott MIT is still of first-order and we cannot expect a smooth collapse of the peak into the anomalous contribution at zero frequency as that reported [5,7] at $T = 0$: On the insulating side of the MIT (right panel), the regular part of the absorption spectrum changes abruptly featuring a large gap with a rather broad (Hubbard-bands-like) bump located at $\omega \sim U$. This bump already starts to develop on the metallic side outside the frequency region displayed in Fig. 4; also note the very different $y$-axis scales. Evidently, as one can easily suppose by comparing the different axis-scales between the two panels of Fig. 4, the regular spectral weight is not the same on both sides of the MIT, reflecting the abrupt appearance of a finite anomalous term ($C > 0$) on the insulating side.

In this context, a reliable quantitative estimate of the value of $C$ can be obtained by exploiting the associated sum rule, which for our case explicitly reads:

$$\pi \chi^T = \int_{-\infty}^{\infty} d\omega \, \frac{\Im \chi^{\mathcal{R}}(\omega)}{\omega} + \pi \beta C \,, \tag{46}$$

which we used to determine $C(\beta, U)$ in this work. Here $\Im \chi^{\mathcal{R}}(\omega)$ came from analytic continuation of $\chi^{\text{th}}(i\omega_n)$ for the positive Matsubara frequencies only (see Eq. (43)) and $\chi^T = \chi^{\text{th}}(i\omega_n = 0)$. Equivalently one can also use:

$$\frac{g^2}{4} \langle \hat{S}_z^2 \rangle = \left( \frac{1}{\pi} \int_{-\infty}^{\infty} d\omega \, \Im \chi^{\mathcal{R}}(\omega) f_{\text{BE}}(\omega) \right) + C \,, \quad \text{with} \quad f_{\text{BE}}(\epsilon) = \frac{1}{e^{\beta(\epsilon - \mu)} - 1} \,. \tag{47}$$

Our quantitative estimates of the anomalous term $C$, identified as the missing spectral weight [7], is shown in Fig. 5 for the whole path at $T = \frac{1}{200}$. For $U < U_{c1}$, this procedure

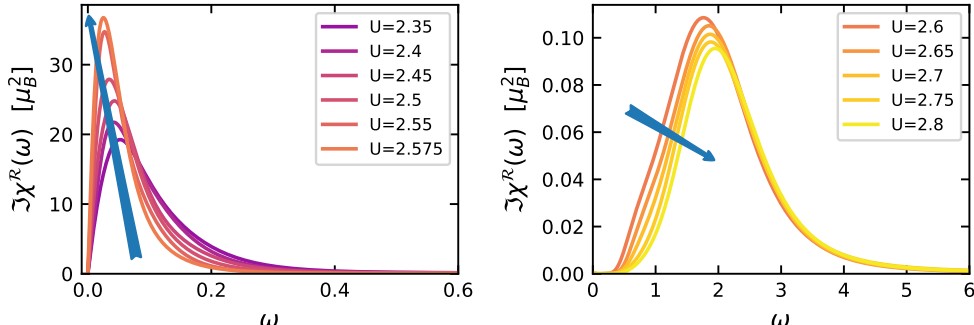

Figure 4: Absorption part of the local spin spectral functions obtained by analytically continuing the DMFT spin-susceptibilities for different $U$ values at $T = 1/200$ in the metallic phase (left) and the insulating phase (right). Notice the different axis-scales used in the two panels.

yields, unsurprisingly $C \cong 0$ as we do expect for a Fermi liquid at low-$T$. In the coexistence region the obtained values for $C$ were finite (although small) in the metallic phase (M→I path). We consider this an artefact of the numerical procedure of calculating C. Indeed, a close inspection of the fit loss Fig. 3 shows that $C = 0$ is within the error bars as is a small value of $C \neq 0$, see Appendix D.

In the Mott insulating phase, at large interaction values ($U > U_{c2}(T = 1/200)$) the local magnetic moment is within 5% of the atomic limit ($C = 1$) reflecting an almost "frozen" spin dynamics [16] . These qualitatively different features are essentially retained by the two-class of solutions within the coexistence region. In particular, for $M \leftarrow I$ one only finds a further slight reduction of the anomalous term and the associated long-term memory effects, with values of $C$ still larger than 0.8 even at $U \sim U_{c1}(T = 1/200)$. At the same $U$, for $M \rightarrow I$, the numerical estimate of $C$, through the missing weight procedure, features very small values. Numerically these are compatible with $C \equiv 0$, which is supported by the close inspection of the fit-loss function previously discussed[11]. A vanishing $C$ at the Mott-transition can also be confirmed from two different methods [7, 55] at $T = 0$, reflecting the appearance of a degenerate groundstate in the MIT.

## 4.4 A map of the coexistence region

Finally, by repeatedly applying the procedure illustrated above for the $T = \frac{1}{200}$ case to several other temperature paths within the coexistence region, we can construct an intensity map of the values of $C$, quantifying the expected long-term-memory effects for the whole parameter region around the first-order MIT. In particular, our results for $C(\beta, U)$ as a function of inverse temperature and interaction strength are shown for $\beta \geq \frac{1}{T_c}$ in the three panels of Fig. 6, for the different cases considered (left: $M \rightarrow I$), (center: $M \leftarrow I$) and (right: thermodynamically stable solutions).

Evidently, the different colors of the intensity plots strongly suggest that at all temperatures within the coexistence region, detectable long-term memory effects can only be found in the insulating solutions. Indeed, this conclusion is quantitatively supported by the more rigorous analysis based on the variation of the blur parameter $b$. In this respect, the different colored lines shown in Fig. 6 correspond, for each temperature, to the $U$ values above which our QMC data can no longer be explained by normal contributions only ($C = 0$) if a given minimal

---

[11]As an additional check we also obtained very accurate data for ($M{\rightarrow}I$); $U = 2.625$; $\beta = 100$ and performed a Padé interpolation as described in Section 3.3, which confirmed that $C \cong 0$ up until the phase transition. (The numerical result for this point in the phase diagram was $C = -0.001$.)

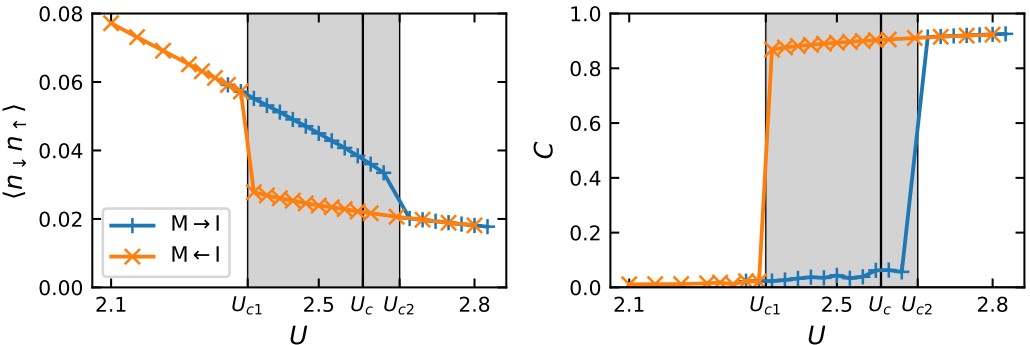

Figure 5: Hysteresis of the double occupancy (left panel, reproduced from Fig. 1) and of the estimated anomalous term in the local spin susceptibility (right) along the chosen $U$-path at fixed $T = 1/200$ across the coexistence region (grey-shadowed area) of the Mott MIT in the half-filled Hubbard model. The black vertical line at $U_c$ [29], which marks the interaction value where the thermodynamic first-order MIT takes place, is defined by the equality condition of the free energies of the two phases.

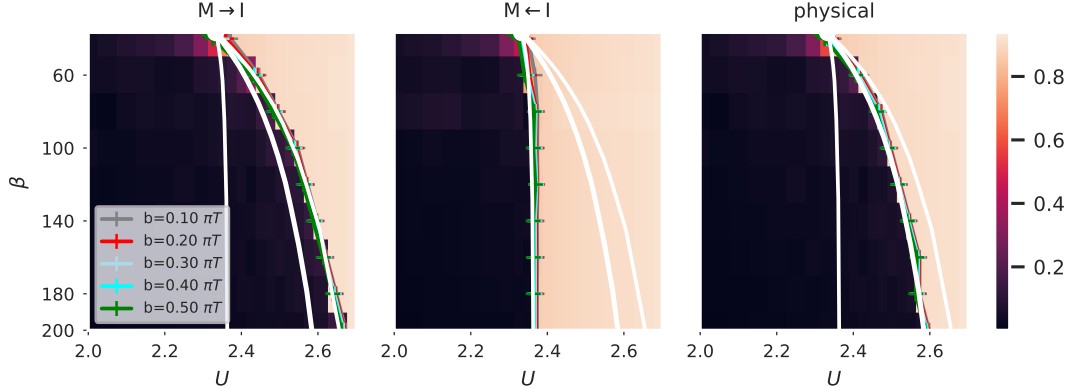

Figure 6: Quantitative estimate of the value of $C$ as a function of $U$ and of the inverse temperature $\beta$ along the whole coexistence region of the Mott MIT. The first two panels encode the results obtained along paths of the kind $M \to I$ (left panel), $M \leftarrow I$ (central panel), while the rightmost panel provides the same estimates for the thermodynamically stable phases. The colored lines show the $(U, \beta)$ tuples at which the fit loss min $\chi^2(b)$ starts to rise steeply. (Marked as squares in the right part of Fig. 11.)

width $b$ for the blurring of the regular part of the spectrum is set. All the lines associated with different values of $b$ shown in the three panels of Fig. 6 collapse to the corresponding transition lines of the calculation considered (respectively: $U_{c2}(T)$, $U_{c1}(T)$ and $U_c(T)$), confirming the emergence of detectable long-term memory effects in the local spin response only after crossing the first-order MIT.

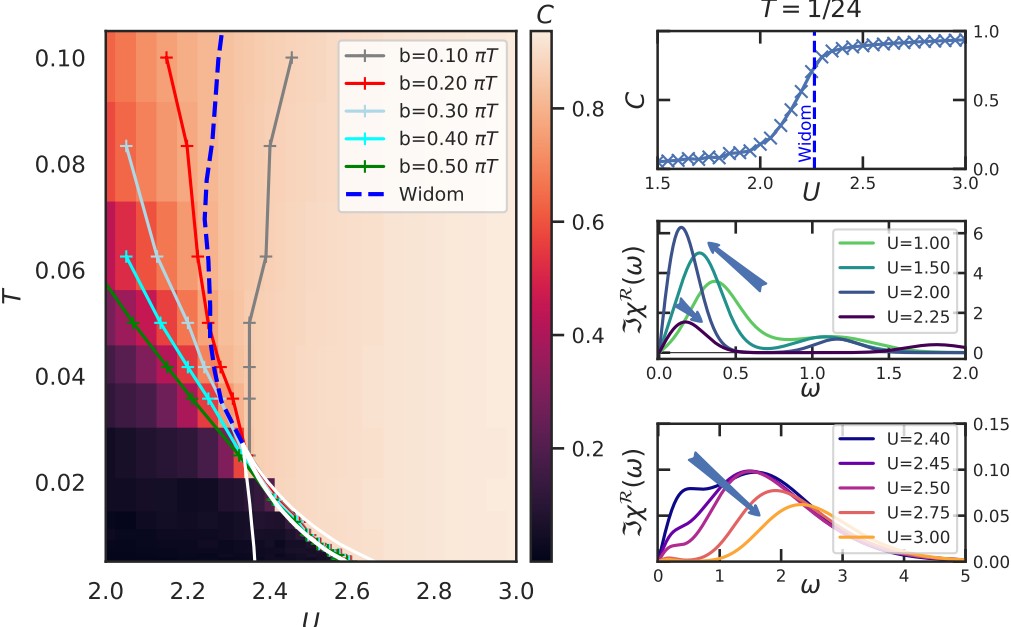

Figure 7: *Left:* Estimate of the anomalous term $C(T, U)$ as a function of temperature and interaction strength over a broader portion of the DMFT phase diagram than the strict Mott MIT regime considered in Fig. 6. Data for the Widom line have been taken from [48]. The other colored curves mark the $T$ and $U$ values above which the QMC data are no longer compatible with only a (sharp) regular low-energy peak, consistent with a blurring of width $b$ (see text). $U_{c1}$, $U_c$ and $U_{c2}$ curves taken from [29]. *Right:* Evolution of the anomalous coefficient $C$ (top panel) as well of the regular part of the absorption spectrum $\Im\chi^{\mathcal{R}}(\omega)$ (central and bottom panels) for increasing values of the interaction $U$ at fixed temperature ($T = 1/24 \simeq 0.042 > T_c$).

# 5 Phase diagram and underlying physics

While the rather sharp behavior of $C$ in the coexistence region appears consistent to the 1st order nature of the Mott-Hubbard MIT, it is interesting to study what happens by raising the temperature above the $T_c$ of the critical point. To this aim, we have repeated the procedure described in the previous section along several paths for $T > T_c$ including a larger interval of $U$ values beyond the extension of the coexistence region.

The obtained results are reported in Fig. 7, whose main panel (on the left side) considerably enlarges the representation of the rightmost part of Fig. 6. As it is apparent from the reported data, the region of significantly large long-term memory effects of local spin correlation does not quickly disappear at, or right above, the critical point of the MIT, but it extends over a large region of the high-$T$ crossover regime. There, however, the overall behavior of $C$ along $U$-paths becomes gradually smoother by further increasing the temperature. On a qualitative level this is evidenced by the progressively milder change of the color tone associated to the magnitude of $C$, found in the high-temperature/crossover region of the phase diagram of Fig. 7. More quantitative information can be gained by inspecting the colored lines, which, similarly as in Fig. 6, mark the $U$ values above which a nonzero value of $C$ can be considered numerically proved for a given blurring threshold $b$. While the lines corresponding to different values of $b$ were essentially collapsing in a single one along the 1st-order MIT, they depart from each other at the critical point of the transition and display a progressively larger spread by further increasing the temperature. Eventually, it is also worth observing that the gradual increase of $C$

through the crossover region appears to stop in the proximity of the coloured line associated to $b = 0.2 \pi T$, after which $C$ reaches plateau values not too different ($C \geq 0.85$) from the atomic limit case[12]. Interestingly, the onset of such an (essentially constant) behavior of $C$ roughly corresponds to the crossing of the so-called Widom line (blue dashed line [48]) in the phase diagram. As already mentioned in Sec. 4, we recall that the Widom line can be regarded a natural prolongation of the true Mott-MIT line at higher temperatures above the critical point.

Our numerical analysis can be further refined by inspecting the spectral properties of the system along one specific path *above* the critical point. To this aim, we performed a similar analysis as that shown for $T = 1/200 = 0.005$ in Figs. 4 and 5 at the higher temperature of $T = 1/24 \simeq 0.042$. The corresponding results are summarized in the right panels of Fig. 7: The upper one shows our quantitative estimate of $C$ via the determination of the missing spectral weight, while in the two other panels we report the corresponding absorption spectra of the local spin susceptibility, computed following the same procedure illustrated in the previous section, for increasing values of $U$ along the selected path. On the basis of the results obtained, it is interesting to note how the smoother increase of the long-memory effects encoded in $C$ corresponds to a somewhat "non-monotonous" evolution of the low energy spectral features of $\Im \chi^{\mathcal{R}}(\omega)$. In particular, for the smallest $U$ values considered in the middle panel of Fig. 7, associated to a still rather coherent electronic behavior, one observes the gradual formation and the progressive softening of a low-energy absorption peak, similarly as in the low-$T$ data of the metallic solutions in the coexistence region (upper panel of Fig. 4). At intermediate (e.g. $U \geq 2$), instead, the softening trend of the lowest absorption peak gets reversed in Fig. 7 (right panels), while this starts displaying a significant broadening. In fact, at high-$T$ it is this latter feature and not the softening, which drives the progressive spectral weight shift from the low-energy sector of the regular spin absorption to the anomalous part responsible of the long-term memory behavior. In the real-time domain this would correspond to a rather quick decay [5] of the local spin-correlation function to the large asymptotic long-term memory term $\beta C$.

The analysis of our DMFT results is suggestive of several physical considerations. In particular, the numerical evaluation of $C$ can be interpreted in the light to the exact Lehmann expression of $C$, reported in Eq. (31) for the general case:

$$
\begin{aligned}
C &= \frac{1}{Z} \sum_{n,m}^{E_n = E_m} e^{-\beta E_n} \left| M_{nm}^z \right|^2 \\
&\equiv \sum_n \frac{e^{-\beta E_n}}{Z} C_n = \sum_n \frac{e^{-\beta(E_n - E_0)}}{\tilde{Z}} C_n \,,
\end{aligned}
\tag{48}
$$

whereas $\tilde{Z} = e^{\beta E_0} Z \simeq N_0 + N_1 e^{-\beta(E_1 - E_0)} + \cdots$ ($N_n$ being the degeneracy of the eigenstate $E_n$). Indeed, Eq. (48) directly links the quantification of the long-term memory effects to intrinsic properties of the underlying (and typically unknown) many-electron energy-spectrum. In particular, if we now consider different temperature-cuts in the phase diagram for fixed values of $U > U_{c2}(T = 0)$, it is clear that the broad plateau of large $C$ values characterizing the entire Mott insulating phase identifies the ground state term, $C_0$, in Eq. (48) as major contribution to the anomalous spin-response of the Mott insulator. At the same time, a finite $C_0$, which is physically consistent with the Curie behavior of the corresponding isothermal susceptibility ($\chi^T \sim \beta C_0$), also implies that the ground state of the full many-electron problem under consideration must be *degenerate*. In fact, consistent to our discussion after Eq. (31), if the ground-state were non-degenerate, $C_0$ in Eq. (48) would reduce to the square of the expectation value of the observable of interest (i.e., $|\langle A \rangle|^2 = |\langle M^z \rangle|^2$), which in our case yields obviously zero.

---

[12]For a concise discussion of the expected trend in high-$T$ regime of the Mott insulating phase, we refer the reader to the Appendix E.

While it is known [2] that the ground-states of Mott-insulating phases computed in DMFT are indeed highly degenerate[13], as also signalled by their large entropy of order $\sim N \ln 2$ [14], it is interesting to underline, here, the intrinsic three-fold relation, encoded in Eq. (48) among (i) the ground-state degeneracy in the Mott phase, (ii) the Curie behavior of the isothermal magnetic response and (iii) long-term memory of local spin correlations in time, i.e. if a local magnetic moment is measured at a given site and then again later after an arbitrarily long time the Mott-insulating system will still largely *remember* the spin-configuration of the first measurement $\lim_{t\to\infty} \langle \hat{M}_z(t)\hat{M}_z(0)\rangle = \frac{C_0}{T} \neq \langle \hat{M}_z\rangle\langle\hat{M}_z\rangle = 0$.

The situation appears rather different for lower interaction values, namely for $U$ less than $U_{c2}(T=0)$. In this case, the observed vanishing of $C$ in the low-$T$ regime, which implies $C_0 = 0$ in Eq. (48), and is clearly consistent with the non-degenerate nature of the underlying Fermi-liquid ground state. At the same time, the appearance of sizable long-term memory effects *above* a certain temperature (to which we will refer as $\bar{T}(U)$) in the crossover regime can be rationalized by postulating the presence of a finite (and large) term ($C_n > 0$) in Eq. (48) associated with *degenerate excited-states* in the many-electron energy spectrum of the lattice model, namely at the energy of $E_n \sim \bar{T}(U) > E_0$. This would indeed correspond to an activated behavior of $C(T) \sim e^{-(E_n-E_0)/T}$. Hence, a plausible interpretation of our results in the whole correlated metallic regime is the following: By increasing $U$, long-term memory effects are linked to excited states of the many-electron spectrum, whose energy difference w.r.t. the (Fermi-Liquid) ground state $E_n - E_0$ tends to decrease with increasing $U$. This suggests that the Hubbard interaction may first drive the formation of (quasi-)degenerate levels at the relatively high energy $E_n$ and, then their progressive descent in the many-electron spectrum. These specific eigenstates could then be regarded as high-energy precursors of the local moment formation in the full many-electron problem. After crossing $U_c(T_c)$, the interaction value corresponding to the critical point, the first order nature of the Mott MIT separates the Hilbert-spaces of the metallic and the insulating phase. As for $U \leq U_{c2}(T=0)$, the ground-state is a non-degenerate Fermi-liquid, $C \approx 0$ in the whole region below the MIT instability line. Above the transition, the grand canonical partition sum over the many-electron eigenstates effectively gets restricted to the insulating ones, which explains the observed jump to large, and essentially temperature independent $C$ values.

The second-order nature of the transition at its two endpoints at zero (for $U = U_{c2}(T=0)$ and finite $T$ (i.e., for $U = U_c(T_c)$) is associated, instead, to a continuous evolution of the physical properties. This is *consistent* to a value of $E_n$ coinciding with the energy of the Mott-insulating state at $U = U_c(T_c)$ and becoming the true ground state of the full-many electron problem at $U \geq U_{c2}(T=0)$.

Eventually, while we have inspected here the case of local magnetic correlations, because of their relevance for the Mott MITs, it is worth emphasizing that long-term memory behaviors might affect, in rather different fashions, the time dependence of other correlations functions. This multifaceted aspect evidently reflects the intrinsic link between the anomalous/long-term memory response and the underlying symmetries of systems, encoded in Eq. (33). For instance, one can easily relate the anomalous response associated to the *total* charge [$\hat{n}^{\text{tot}} = \sum_{i,\sigma} \hat{c}^\dagger_{i\sigma}\hat{c}_{i\sigma}$] and/or the *total* spin [e.g., $\hat{S}^{\text{tot}}_z = \frac{1}{2}\sum_{i,\sigma}(\hat{c}^\dagger_{i\uparrow}\hat{c}_{i\uparrow} - \hat{c}^\dagger_{i\downarrow}\hat{c}_{i\downarrow})$] observables, which are conserved quantities for several non-magnetic many-electron Hamiltonians,

---

[13]This is intuitively understood as in the Mott phase each lattice-site is only occupied (approximately) by a single electron of spin 1/2 as double-occupancies are energetically unfavorable. The formed local moments are, however, randomly oriented with respect to one another at different sites.

[14]We briefly recall here that the ground-state degeneration of the Mott insulating phase, as well the apparent violation of third thermodynamic principle related to the non-vanishing $T \to 0$ entropy, are automatically resolved in DMFT by the spontaneous breaking of the SU(2)-symmetry associated to the onset of AF-long range order. As we only consider here the case of *paramagnetic* DMFT calculations, we will not explicitly discuss further this aspect, which anyway does not affect our general considerations on the many-electron energy-spectrum.

to the well-known jump of the corresponding charge/magnetic uniform responses between the *static* ($\mathbf{q} \to 0, i\omega_n \equiv 0$) and the *dynamic* ($\mathbf{q} \equiv 0, i\omega_n \to 0$) limit, yielding respectively $\chi^T_{\mathbf{q}=0}$ and $\chi^{\mathcal{R}}_{\mathbf{q}=0}(\omega = 0)$, whose relevance has been recently discussed also in a DMFT context [56]. Analogous discontinuities in the zero-frequency limit of the $\mathbf{q} = 0$ spin response have been recently noted [53] also in DMFT calculations of the antiferromagnetically ordered Hubbard model in presence/absence of a external uniform magnetic field.

Evidently, the link encoded in Eq. (31) between the anomalous part of a response function and intrinsic properties of the energy spectrum will generally hold *independently* of the observable considered. Hence, analyzing the long-term memory effects of different response functions might allow to gain complementary insightful information about the underlying eigenstate structure of the many-electron problem.

# 6 Conclusions

We have investigated the multifaceted algorithmic and physical implications of an anomalous term ($C$) in the response functions of interacting electron systems. Its presence corresponds to a jump-discontinuity between the static isothermal susceptibility and zero-frequency limit of the dynamical Kubo response function, directly reflecting the emergence of a non-decaying behavior of the corresponding correlations in the real time domain ("long-term memory" effect). This phenomenon is formally linked to the underlying symmetries of the many-electron Hamiltonian, and it can be shown that under certain conditions, a finite value of $C$ directly reflects the existence of degeneracies in the many-particle energy spectrum. Through $C$ and its temperature evolution we can thus gain some indirect information on the many-particle spectral density.

The algorithmic procedure, we presented in Sec. III of our paper, has been thus designed for reliably detecting the presence of anomalous contributions in a given many-electron response and to evaluate their size. The performance of the proposed scheme has been then successfully verified in Sec. IV-V by hands of its application to a DMFT-based analysis of the long-term memory features of the local magnetic response across the Mott MIT of the Hubbard model.

Beyond the interesting insights gained on temporal and spectral aspects of this testbed problem, our scheme is directly applicable to any kind of response function of many-electron systems. From an algorithmic point of view, a reliable estimate of the value of $C$ might considerably help a subsequent analytic continuation for the regular part of the system response, improving the quality of the calculated absorption spectra. In the most challenging intermediate-coupling regimes, in particular, a preliminary estimate of $C$ might represent an essential prerequisite for an accurate analytic continuation. At the same time, the study of long-term memory effects in different response functions can yield complementary new pieces of information about the fundamental properties of the many-electron Hamiltonians under investigation, possibly useful also in perspective of experimental observations made beyond the thermodynamic equilibrium.

# Acknowledgements

We would like to thank L. Del Re, A. Kauch, J. Kaufmann, F. Krien, J. Kuneš, E. Moghadas, G. Rohringer, G. Sangiovanni, J. Tomczak, E. van Loon and M. Wallerberger for several exchanges of ideas.

**Funding information** This work was supported by the Austrian Science Fund (FWF) through projects P 30819 (CW), P 32044 (KH), and I 2794-N35 (AT). Calculations have been done on the Vienna Scientific Cluster (VSC).

# A   Considerations on nonlinear response

Similar complications as those arising for the bosonic two-point correlation function at zero frequency also appear for three- or more general $l-$point functions. In this respect a spectral representation, including anomalous terms, was to our knowledge originally published in [57,58] for the three and four point functions. Recently Kugler et al. [27] gave an elegant description for the $l-$point functions.

These formal results have a direct relation with *nonlinear response theory* (NLR). As Kubo already remarked [18] his formalism is not limited to the first order term in the external field. If the full time-dependent Hamiltonian is given by

$$\hat{\mathcal{H}}(t) = \hat{H} - \mathcal{F}_{(t)}\hat{A}, \tag{49}$$

where $\mathcal{F}_{(t)}$ is an external (classical) field, which is zero for $t < 0$, the fluctuations in the expectation value of any system observable $\langle \hat{B}(t) \rangle$ are given by

$$
\begin{aligned}
\langle \delta \hat{B}_{(t)} \rangle &= \langle \hat{B}_{(t)} \rangle_{\mathcal{F}} - \langle \hat{B}_{(t)} \rangle_{\mathcal{F}=0} \\
&= \sum_{l=1}^{\infty} \int_{-\infty}^{t} dt_1 \int_{-\infty}^{t_1} dt_2 ... \int_{-\infty}^{t_{l-1}} dt_l \\
&\quad \cdot i^l \left\langle \left[ \left[ ... \left[ \hat{B}_{(t)}, \hat{A}_{(t_1)} \right], \hat{A}_{(t_2)} \right], ..., \hat{A}_{(t_l)} \right] \right\rangle_0 \\
&\quad \cdot \mathcal{F}_{(t_2)} \mathcal{F}_{(t_2)} ... \mathcal{F}_{(t_l)}.
\end{aligned}
\tag{50}
$$

Absorbing the integration boundaries into the commutator of commutators defines a NLR susceptibility. The $l$th order contribution in $F$ is given by

$$
\begin{aligned}
\langle \delta \hat{B}_{(t)} \rangle^{(l)} &\equiv \left( \prod_{i=1}^{l} \int_{-\infty}^{\infty} dt_i \, \mathcal{F}_{(t_i)} \right) \chi^{BA^l \, \mathcal{R}}_{(t,t_1,t_2,...,t_l)} \\
&= \left( \prod_{i=1}^{l} \int_{-\infty}^{\infty} dt_i \, \mathcal{F}_{(t_i+t)} \right) \chi^{BA^l \, \mathcal{R}}_{(0,t_1,t_2,...,t_l)},
\end{aligned}
\tag{51}
$$

where time-translation in-variance has been used in the second line. Fourier-transforming gives

$$
\langle \delta \hat{B}_{(\omega)} \rangle^{(l)} = \frac{1}{(2\pi)^{l-1}} \left( \prod_{i=1}^{l} \int_{-\infty}^{\infty} d\omega_i \, \mathcal{F}(\omega_i) \right) \chi^{BA^l \, \mathcal{R}}_{(0,-\omega_1,-\omega_2,...,-\omega_l)} \delta(\omega - \textstyle\sum_{i=1}^{l} \omega_i),
\tag{52}
$$

which evidently corresponds to the generation of higher harmonics. [59] showed, quite elegantly, that there is a simple relation to the corresponding $l + 1$-point thermal susceptibility:

$$
\begin{aligned}
\chi^{\text{th}}_{BA^l}(\tau_1, \tau_2, ..., \tau_l) &= \frac{(-1)^l}{l!} \langle \mathcal{T} \hat{A}_l(-\tau_l)...\hat{A}_2(-\tau_2)\hat{A}_1(-\tau_1)\hat{B}(0) \rangle, \\
\chi^{\text{th}}_{BA^l}(i\omega_{n1}, i\omega_{n2}, ..., i\omega_{nl}) &= \frac{1}{\beta^l} \int_0^{\beta} ... \int_0^{\beta} d\tau_1 \, d\tau_2 ... d\tau_l \, e^{i\tau_1 \omega_{n1} + i\tau_2 \omega_{n1} + ... + i\tau_l \omega_{nl}} \\
&\quad \cdot \chi^{\text{th}}_{BA^l}(\tau_1, \tau_2, ..., \tau_l).
\end{aligned}
\tag{53}
$$

One simply has to replace all Matsubara frequencies in Eq. (53) by the corresponding real frequencies plus the *same* infinitesimal imaginary shift $\delta \to 0^+$:

$$
\chi^{BA^l \, \mathcal{R}}_{(0,\omega_1,...,\omega_l)} = \chi^{\text{th}}_{BA^l}(i\omega_{n1} \to \omega_1 + i\delta, ..., i\omega_{nl} \to \omega_l + i\delta).
\tag{54}
$$

However, it appears that it was implicitly assumed in their derivation that no degeneracies are present for the zeroth Matsubara frequency/-ies (see [59, Eq. 32]). Naturally, this excludes all anomalous terms. In real time/frequencies no such assumption was made. Hence, their results in real frequencies can be considered general. In Eq. (54) one should use for the right hand side only the regular contributions. (In analogy to Eq. (27).) This is also reasonable when comparing it to the two-point function, (or linear response), case. Indeed, the commutator exactly removes the anomalous term which only contributes to the anti-commutator but not to a commutator of commutators as in Eq. (50).

## B Computational details

**Dynamical Mean Field Theory –**    The DMFT simulations were performed with a continuous-time quantum Monte Carlo (QMC) algorithm implemented in the code package `w2dynamics` [35]. For the convergence of the self-consistency cycle worm-sampling with symmetric-improved estimators [36] was employed `SelfEnergy=symmetric_improved_worm`. As a convergence criterion a Hotelling test [60] of the self-energy of consecutive DMFT iterations on the first 20 Matsubara frequencies was used. Additionally the results were checked by visual inspection considering the last 5 iterations. Depending on the parameters 20 to 150 DMFT iterations were necessary for the convergence. (e.g. close to the phase transition in the coexistence region more iterations were necessary.) For computing the one-particle quantities we used `Nmeas=`$5 \cdot 10^5$ to $10^7$ QMC measurements (depending on temperature) on each of the 64 cores, where the calculation was done in parallel. Particle-hole symmetry was obtained by choosing a chemical potential equal to the Hartree term $\mu = U/2$.

On top of the converged DMFT-solution we then calculated with a single statistic-step (fixed one-particle quantities) the thermal susceptibility $\chi^{\text{th}}(\tau)$ with the segment solver. The calculation was again done in parallel with 64 cores. For `Nmeas` $10^5$ was used $T > 1/10$ and $10^6$ otherwise.

## C Approximate analytic forms for $\chi^{\mathcal{R}}$

Several analytic forms for the regular term are possible. A particularly simple expression with Lorentzian shape was suggested in [61, Eq. (2)], where it was successfully applied to study the effective local magnetic moment in $\alpha$- and $\gamma$-iron. The proposed function reads

$$\chi_L^{\mathcal{R}}(\omega) = A\frac{i\delta}{\omega+i\delta} = A\frac{\delta^2}{\omega^2+\delta^2} + iA\omega\frac{\delta}{\omega^2+\delta^2}, \tag{55}$$

where $A = \frac{\mu_{\text{eff}}^2}{3T}$ in the local moment regime [61]. It is interesting to note [61] that this heuristic form can be used to capture the anomalous terms by taking the $\delta \to 0$ limit [15]. In particular, one can derive all our expressions involving $C$ by separating the Kubo susceptibility into $\chi^{\mathcal{R}}(\omega) = A\frac{i\delta}{\omega+i\delta} + \chi_{\text{reg.}}^{\mathcal{R}}(\omega)$. For instance, our form of the fluctuation dissipation theorem (for $\hat{A} = \hat{B}$) can be re-derived as

$$
\begin{aligned}
\Psi_{AA}(\omega) &= 2i\Im\chi^{\mathcal{R}}(\omega)\coth(\beta\omega/2) \\
&= 2iA\delta\frac{\omega}{\omega^2+\delta^2}(\frac{1}{\beta/2\omega} + \mathcal{O}(\beta\omega)) + 2i\Im\chi_{\text{reg.}}^{\mathcal{R}}(\omega)\coth(\beta/2\omega) \\
&\stackrel{\delta\to0^+}{=} 4\pi iA/\beta\ \delta(\omega) + 2i\Im\chi_{\text{reg.}}^{\mathcal{R}}(\omega)\coth(\beta/2\omega),
\end{aligned}
$$

where one identifies $A = C\beta$ and $C = \mu_{\text{eff}}^2/3$.

---

[15]See also our answer to point 4 of the Referee report (A) which is available online.

Table 1: Fit-parameters of $\chi_{\text{HO}}^{\text{th}}(i\omega_n)$ for selected $U$-values at $\beta = 200$.

| $U$ | fit parameters | | |
| --- | --- | --- | --- |
| | $\omega_0$ | $\gamma$ | $A$ |
| 2.35 | 0.093 | 0.100 | 0.259 |
| 2.55 | 0.051 | 0.053 | 0.137 |

At the same time, it should be stressed that, in the framework of our calculations, the simplified form of Eq. (55) (with a finite $\delta$) could not be applied to express the *regular* part of the magnetic response. Indeed, fitting the analytic continuation (to the upper complex half-plane) of Eq. (55): $\chi_L^{\text{th}}(\omega_n) = A\delta/(|\omega_n| + \delta)$ to the QMC data did not yield a good fit for our case (not shown). One possible reason is that the boundary condition for $t = 0^+$ is not satisfied by Eq. (55). More specifically, from the definition in Eq. (9) is is clear that (i) $\chi_{AB}^{\mathcal{R}}(t) \in \mathbb{R}$ (ii) $\chi_{AA}^{\mathcal{R}}(t \to 0^+) \propto \langle[\hat{A}(t \to 0^+),\hat{A}]\rangle = 0^+$. (iii) $\chi_{AB}^{\mathcal{R}}(t < 0) = 0$. Property (iii) is equivalent to the fact that $\chi^{\mathcal{R}}(\omega)$ has no poles in the upper complex half plane.

It should be noticed that properties (i) + (ii) cannot be fulfilled simultaneously by a $\chi^{\mathcal{R}}(\omega)$ that has only a single (first order) pole in the lower complex half-plane. For instance, the Fourier-transform of Eq. (55) is: $\chi_L^{\mathcal{R}}(t) = A\delta\,\theta(t)e^{-\delta t}$. In frequencies one needs a function with at least two (first order) poles[16] in the lower complex half plane.

It should, however, be noted that (ii) assumes $\hat{A} = \hat{B}$. For cases where the two operators are different (ii) will be in general not fulfilled. This might allow for a broader applicability of Eq. (55).

A particularly simple example of an expression guaranteeing two poles in the lower complex half plane is given by the mathematical form of the fundamental solution to the harmonic-oscillator differential equation: $\left(\partial_t^2 + 2\gamma\partial_t + \omega_0^2\right)\chi_{\text{HO}}^{R}(t) = A\delta(t)$ which reads

$$\chi_{\text{HO}}^{\mathcal{R}}(\omega) = \frac{A}{-\omega^2 - 2i\gamma\omega + \omega_0^2} \tag{56}$$

$$= -A\frac{1}{\omega - \Omega_+}\frac{1}{\omega - \Omega_-}, \quad \text{with} \quad \Omega_\pm = -i\gamma \pm \sqrt{\omega_0^2 - \gamma^2},$$

$$\chi_{\text{HO}}^{\mathcal{R}}(t) = A\theta(t)\frac{i}{\Omega_+ - \Omega_-}\left(e^{-i\Omega_+ t} - e^{-i\Omega_- t}\right), \tag{57}$$

$$\chi_{\text{HO}}^{\text{th}}(i\omega_n) = \frac{A}{\omega_n^2 + 2\gamma|\omega_n| + \omega_0^2}. \tag{58}$$

Evidently this expression fulfills the properties (i)-(iii). This phenomenological model was actually used by some of us to analyze the timescales of spin-dynamics for prototypical Fe-based superconductors in the paramagnetic phase [10]. In fact, it can also be used as an approximate expression of the regular term in the present case. In Fig. 8 we show the QMC-data compared to a fit, which was performed for the first 80 positive Matsubara frequency points by minimizing the least-square deviation of Eq. (58). The fit parameters are given in Table 1. We conclude that in this parameter regime[17] spin-fluctuations can be described reasonably well by this simplified expression.

---

[16]or one second order pole

[17]See [62, Ch. 4] for a more detailed analysis at larger temperatures as well as in the insulating phase.

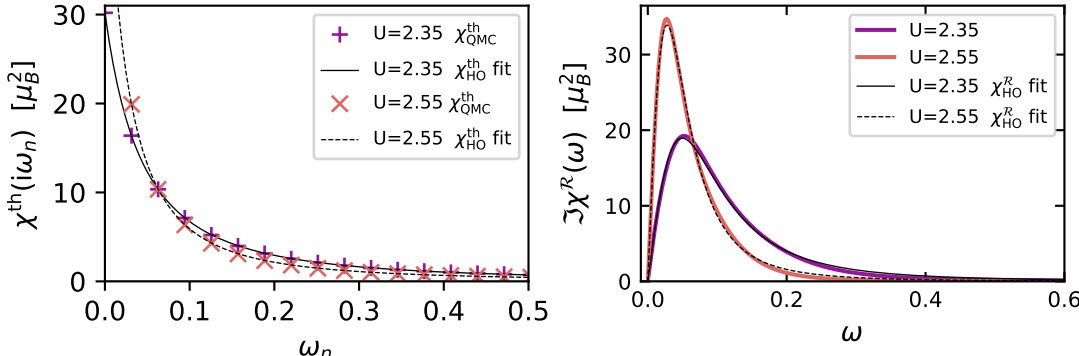

Figure 8: *Left:* Comparison of QMC-data with $\chi^{\text{th}}_{\text{HO}}$ for the fit-parameters of Table 1. *Right:* MaxEnt results compared to evaluating Eq. (56) for the fit parameters obtained from fitting Eq. (58).

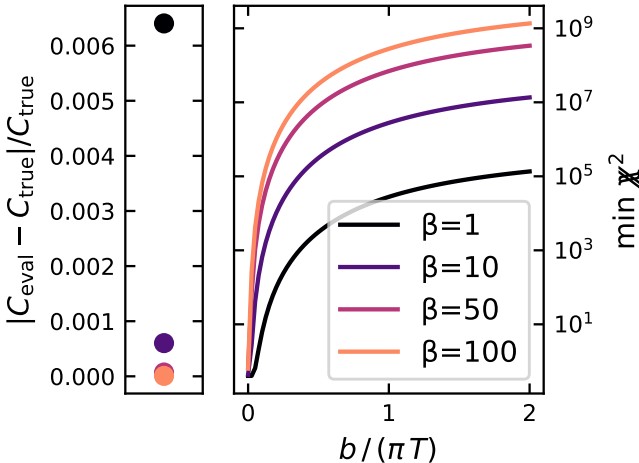

Figure 9: Test of the extraction method with "artificial" data for the exactly known atomic limit case. Relative error in the predicted $C$ (left); Minimal value fit loss $\mathcal{X}^2$ as a function of blur-width $b$ for different temperatures (right).

# D Details on the minimal allowed peak width and simple error estimate

**Test case atomic limit:** To double-check our method for extracting the maximally allowed peak width compatible with the QMC data (see discussion before and after Eq. (45)) we tested it for the atomic limit. The artificial test data was generated by using the analytic result $\chi^{\text{th}}(i\omega_n) = \delta_{n,0}\beta\langle\hat{M}_z^2\rangle$ and adding some white noise for the higher Matsubara frequencies with magnitude $10^{-4}$. Continuing the positive frequencies and comparing the result to $\chi^{\text{th}}(i\omega_n = 0)$ has yielded a very accurate prediction for $C$. As expected the problem gets harder for higher temperatures: Increasing $T$ leads to a larger relative error for $C$; see left part of Fig. 9. A relative error of less than 1% is, however, good enough for most purposes.

**Test case harmonic oscillator:** It is expected that the error of estimating $C$ does not only depend on the temperature but also on the regular signal from which $C$ needs to be distinguished.

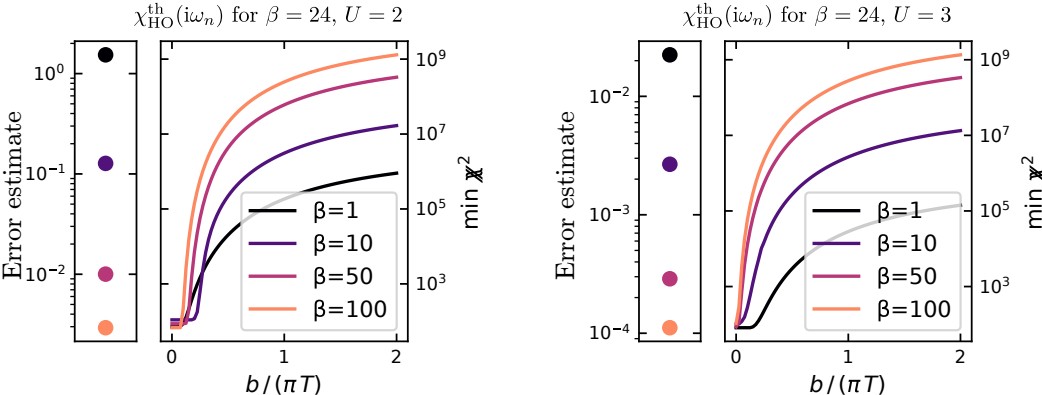

Figure 10: Harmonic oscillator test case. Error estimate and fit loss vs. blur width. *Left:* HO parameters corresponding to $\beta = 24$, $U = 2$ *Right:* HO parameters corresponding to $\beta = 24$, $U = 3$.

More realistic test data than for the atomic limit can be generated by exploiting the harmonic-oscillator formulas Eqs. (56) to (58) as a regular contribution in addition to an anomalous part ($C \neq 0$). For $(A, \omega_0, \gamma)$ we considered two test cases corresponding to fitting QMC data for $\beta = 24$, $U = 2.0$ and $\beta = 24$, $U = 3.0$. The former leads to $\omega_0 = 0.18$, $\gamma = 0.27$, $A = 0.40$ while the later gives $\omega_0 = 2.45$, $\gamma = 0.33$, $A = 0.19$. For both cases we then added Gaussian white noise with standard-deviation of $10^{-3}$. This noise amplitude is rather large, but realistic[18].

In Fig. 10 the resulting error estimations are shown for both cases. The reason for the large error estimate for the left part of Fig. 10 is that in this case the width of the regular part becomes comparable to the temperature. In particular, close to $M \to I$ phase transition, where the preformed local moment is signaled by a narrow (regular) contribution, an accurate prediction for $C$ may become quite challenging. Indeed, this explains why in Fig. 5 a small but finite value for $C$ is estimated for $M \to I$ close to the phase transition, although a closer analysis of the fit loss showed that $C \approx 0$ in this regime: The finite estimated value is an artifact of the numerical procedure in a situation where the first peak in the regular part is very sharp and located at very low frequencies. In fact, a similar analysis as the one presented in Fig. 10 showed that for $M \to I$ $U = 2.6$, $\beta = 200$ the error estimate is of the order of the estimated value for $C = 0.06 \pm 0.05$ (not shown).

Eventually, we should also note that other factors which were not considered for our generated test data (e.g. no white noise due to off-diagonal covariance matrix) might further increase the error estimate.

**Detailed analysis for T=const. and U=const.:**  In the left part of Fig. 11 we show the fit loss as a function of temperature in the insulating phase. Demanding even a very small value for the minimal peak width leads to a large increase in the fit loss. This behavior does not depend strongly on temperature for the range of temperatures we considered. We conclude that in the insulating phase the anomalous term is the most reasonable explanation of our numerical data. Its magnitude does not depend strongly on the temperature for $U = 3$ and is roughly C=0.94. As discussed in section V, we interpret the temperature-independence of $C$ as a manifestation of the ground-state degeneracy.

In the right part of Fig. 11 we show the minimal value of the fit loss ($\chi^2$) as a function of blur width for T=1/24. Which is larger than the temperature $T_c = 0.027$ of the critical point.

---

[18]It is expected that a systematic way of decreasing the error of an analytic continuation is to obtain better data with less QMC-noise. This is however not always possible, especially for a multi-orbital calculations.

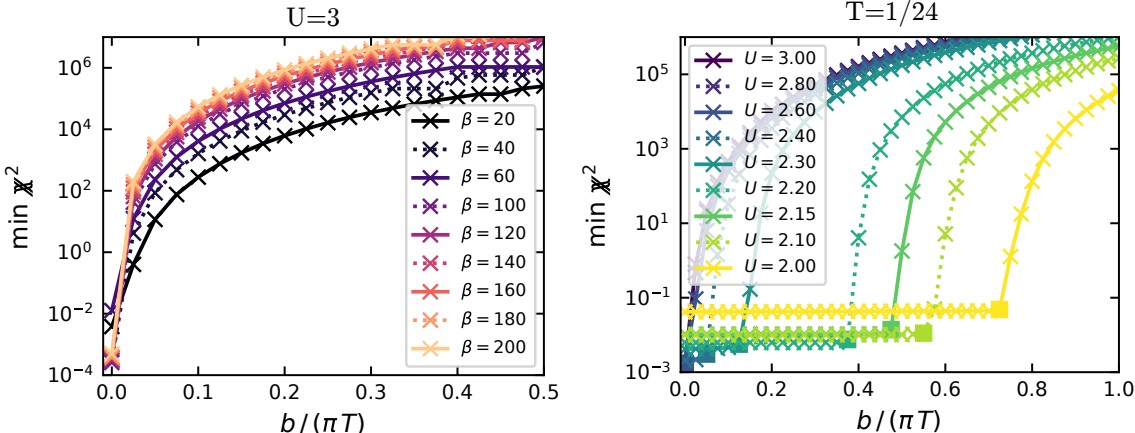

Figure 11: *Left:* various temperatures in the insulating phase with the onsite interaction $U = 3$. *Right:* Minimum value of the fit loss for a range of interaction values at a temperature above the critical point $T = 1/24$.

Changing the interaction value $U$ trough the crossover region shows that differently than in the Mott insulator region our results depend strongly on $U$. After a certain value of the chosen blur width we are no longer able to obtain a good fit. These values (marked as squares in Fig. 11) are the basis for the additional lines we showed in the phase diagram in Section 4.

## E  On the high temperature limit for $C$

Equation (31) can be simplified in the large temperature limit. The asymptotic value of $C$ for $\beta \to 0$ (which practically corresponds to temperatures much larger than *all* other relevant energy scales of the systems) reduces to the following expression:

$$C(\beta) = \frac{1}{Z} \sum_{\substack{l,m \\ E_l = E_m}} e^{-\beta E_l} |A_{lm}|^2 \overset{\beta \to 0}{=} \frac{1}{\dim H} \sum_{\substack{l,m \\ E_l = E_m}} |A_{lm}|^2, \tag{59}$$

where we considered the case with $\hat{A} = \hat{B}$ and $\langle A \rangle = 0$, $A_{lm} = \langle l|\hat{A}|m\rangle$, and $\dim H$, i.e., the dimension of $\hat{H}$, defines the partition function in the large temperature limit $[Z(\beta \to 0)]$. Hence, in this regime, the corresponding value of $C$ is simply given by the sum over all degenerate states, weighted with the corresponding matrix elements.

It is however important to remark that, at very large temperatures, finite values of $C$ will not have any relevant consequence for susceptibility-measurements in different experimental setups, as

$$\chi^T - \chi^{\mathcal{R}}(\omega = 0) = \beta C, \tag{60}$$

so that we find in the large temperature limit for any finite $C > 0$:

$$\chi^{\mathcal{R}}(\omega = 0) = \chi^S = \chi^T. \tag{61}$$

As a pertinent example, one can consider the atomic limit at half-filling. Here, for $\hat{A} = \hat{B} = \hat{M}_z$ one indeed obtains a finite value of $C(T) = \frac{1}{e^{-\beta U/2} + 1} > 0$ at all temperature, with a non-vanishing high-$T$ asymptotic value of $\frac{1}{2}$. The full temperature dependence is shown in Fig. 12.

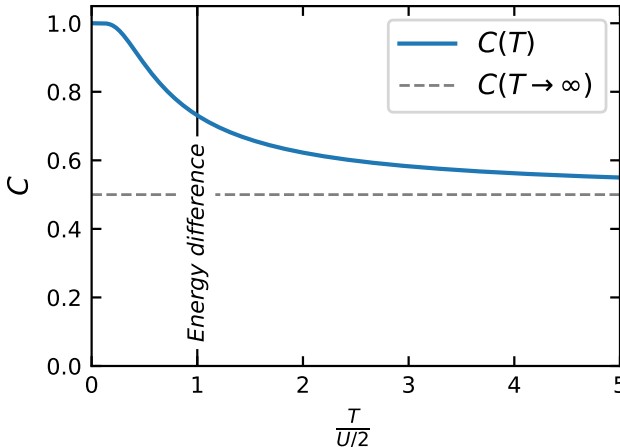

Figure 12: Temperature dependence of the long-term memory coefficient $C$ for the local spin-susceptibility in the atomic limit for $\mu = U/2$.

Consistent with the discussion above, the finite value of $C(\beta = 0) = \frac{1}{2}$ does *not* reflect a finite difference the between $\chi^T$ and $\chi^{\mathcal{R}}(\omega = 0)$ in the $\beta \to 0$ limit, as one finds $\chi^T \simeq \beta$ and $\chi^{\mathcal{R}}(\omega = 0) = 0$.

A final remark is due about the relation of the high-$T$ results for the atomic limit, where $C(\beta \to \infty) = \frac{1}{2}$ and the DMFT calculations in the Mott phase shown at the end of Sec. 5. In this respect, it should be stressed that, as clearly illustrated in Fig. 12, the asymptotic high-$T$ value of C sets in only for $T \gg \frac{U}{2}$, i.e., at temperatures larger than the characteristic energy gap of the half-filled atomic limit problem ($\Delta E = \frac{U}{2}$). Hence, while one might expect to find $C \simeq \frac{1}{2}$ in the high-$T$ regimes of both the Mott phase computed in DMFT and the atomic limit, this only holds at much higher temperatures than those characterizing the Mott MIT itself.

In particular, the largest temperature considered in the phase diagrams of Sec. 5 is $T = 0.1 \ll \frac{U}{2}$. Hence, we are still much closer here to the $T = 0$ regime of the atomic limit (where $C \simeq 1$), than to its $T \to \infty$ regime. For this reason, in the Mott phase of DMFT we only observe a slight decrease of $C(T)$ by increasing the temperature up to $T = 0.1$.

# F  Additional results

It was not the purpose of this work to redetermine the Mott metal-insulator phase transition with high accuracy. This was already done in previous work [29]. Figure 13, however, confirms that our data is in reasonably good agreement with earlier works. The error-bars show the distance to the next $U-$value for which a calculation was preformed by us. As a handy criterion to check whether a DMFT solution belongs to the insulation or the metallic phase we used the ration between the first two values of the self energy

$$\frac{\Sigma(i\omega_n = i\pi T)}{\Sigma(i\omega_n = i3\pi T)} \lessgtr 1,$$

which at low temperature yields a simple, reasonably reliable estimate for the parameters in the phase diagram at which the single-particle scattering-rate becomes significantly enhanced close to the Fermi-surface.

Figures 6 and 7 suffer from the common problem that such a heat map lacks quantitative information and the color bar may mislead the eye. Hence we also plot in Fig. 14 the actual values of $C(\beta, U)$ obtained in our calculation.

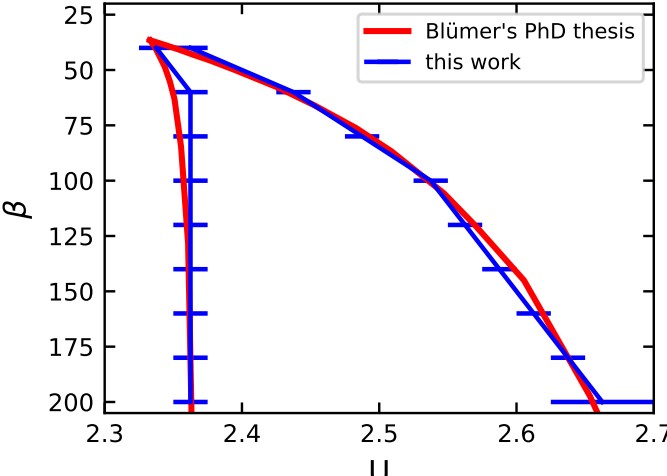

Figure 13: Comparison of $U_{c1}(T)$ and $U_{c2}(T)$ obtained by us compared to the available values in the literature [29] Our error-bars give the distance to the next $U-$values for which a calculation was done.

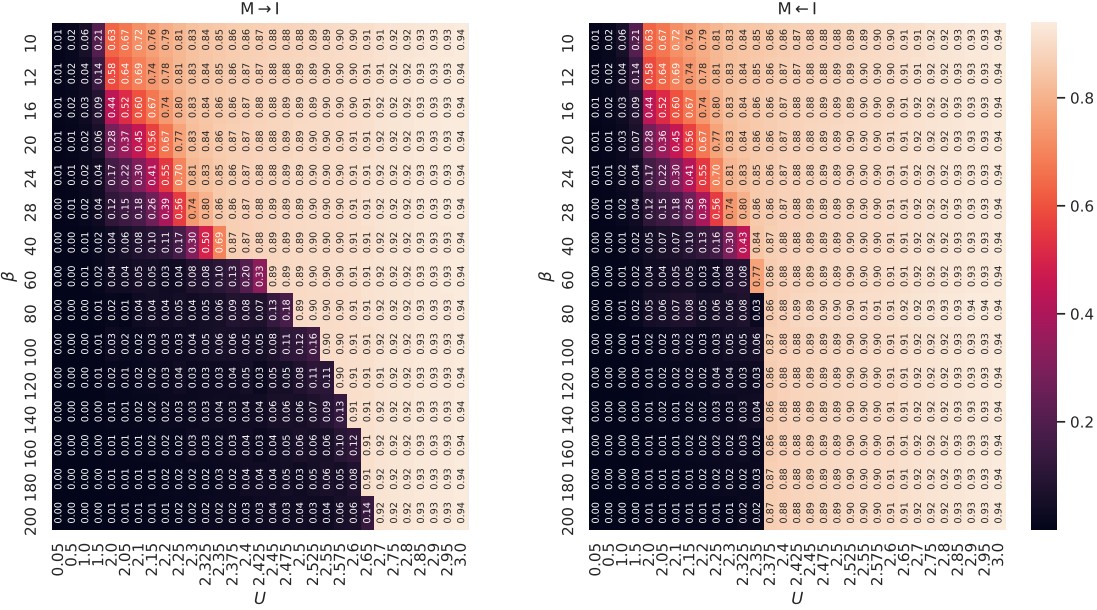

Figure 14: Annotated estimation of $C(\beta, U)$ of our DMFT calculations. In the insulating phase $C$ is almost independent of temperature. This shows that for large interaction values $U > U_{c2}(T = 0)$ $C \neq 0$ is due to a degeneracy of the Mott-insulating ground state (see Section 5 and [7, 55]).

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
