# Peer review of "Long-term memory magnetic correlations in the Hubbard model: A dynamical mean-field theory analysis"

_SciPost Physics, doi:SciPost Phys. 12, 184 (2022)_

## Round 1 · Referee Report · Andrey Katanin (Referee 1) · 2022-1-4

Strengths

The paper studies long-time contributions to the susceptibility (e.g. spin susceptibility) and investigates important example of the vicinity of Mott transition in the single-band Hubbard model

Weaknesses

The suggestions below have to be considered; some corrections are required

Report

The paper by C. Watzenboeck studies long-time contributions to the susceptibility (e.g. spin susceptibility). The important case of long-time contributions to local spin susceptibility of single-band Hubbard model is considered. In general the paper is well-written, and sheds new light on the long-time contributions to susceptibilities in general. The study of such contribution to local spin susceptibility of single-band Hubbard model allows deeper understand formation of local magnetic moments.

Requested changes

  1. The authors use a notion of isolated (Kubo) susceptibility, which is also called by them sometimes isolated (Kubo-Nakano) susceptibility in the abstract and main text of the paper. I am not sure that this notion is rather common (in particular, in what sense the susceptibility (or system) is isolated?). I would suggest to name this quantity as a static limit of the retarded spin correlation function, or somehow else.

  2. In the second and third paragraph of the Introduction I think it is important to mention everywhere that the authors talk about slowing-down of local spin fluctuations ("local" is mentioned only once in the beginning of second paragraph, but it is preferable I think to insert it further, since this is crucial to differentiate this from the critical slowing down etc.).

  3. Regarding this "local" slowing down, I think it is important to cite the paper 10.1103/PhysRevLett.101.166405 which was one of the first papers which discussed this topic for Hund metals.

  4. Regarding Eqs. (25), (28), as well as Eqs. (19), (20), I would like to emphasize that the "anomalous" C-terms can be viewed as a certain limit of some regular contribution to the susceptibility. One such phenomenological form of the regular part, which provides these anomalous terms, was suggested in Eq. (2) of Ref. 10.1103/PhysRevB.88.155120, and reads in the notations of the author's paper chi^R(w)=C\beta i\delta/(w+i\delta), where \delta->+0. Then Im(\chi^R(w))=C\beta w \delta/(w^2+\delta^2), Re(\chi^R(w))=C\beta \delta^2/(w^2+\delta^2), chi^{th}(iw_n)=C\beta\delta/(w_n+\delta)->C \beta \delta_{n,0} at \delta->0.

  5. It would be good in my opinion to clarify how the first order transition line in Fig. 1 is determined. Is it obtained from the comparison of full energies of metallic and insulating solutions, or their free energies, or something else?

  6. The definition of the Widom line is in my opinion better present in the beginning of Sect. 4 (first line of page 14).

  7. In Sect. 4.1 the authors discuss "rapid decay" of local spin correlation function in the insulating phase; I think it is better to characterise it as a rapid decrease, since change in the magnitude of the correlation function is not large in that regime.

  8. For the data presented in the right part of Fig. 2 and left part of Fig. 4 the authors might consider the comparison to the analytic forms discussed above in p. 4 and/or suggest their own analytic forms.

  9. Regarding right part of Fig. 5: do I understand correctly that finite (although small) C in the metallic phase (M->I path) is an artefact of the numerical procedure of calculating C? If yes, I think it is worth to write this explicitly.

  10. In the upper plot of the right part of Fig. 7 I think it is worth to mark the position of the Widom line.

  • validity: top
  • significance: high
  • originality: high
  • clarity: top
  • formatting: perfect
  • grammar: perfect

Author:  Clemens Watzenböck  on 2022-04-05  [id 2358]

(in reply to Report 1 by Andrey Katanin on 2022-01-04)
Category:
answer to question

We thank the Referee for the careful reading of our manuscript, for his appreciation of our work and for finding our results suited for being presented in SciPost Physics.
The Referee has made constructive and useful comments/observations in his report. They have helped us improving the clarity and the precision of our presentation as well as the completeness of our bibliography. A structured reply to the observations of the Referee is attached as *"reply_and_latexdiff_LTM.pdf"*

Attachment:

reply_and_latexdiff_LTM_FkoAc7S.pdf

---

## Round 1 · Referee Report · Anonymous (Referee 2) · 2022-1-12

Report

The manuscript "Long-term memory magnetic correlations in the Hubbard model: A dynamical mean-field theory analysis" analyzes the existence of non-decaying spin correlations in the Hubbard model in the vicinity of the Mott transition. Using DMFT with Quantum Monte Carlo, the authors analyze the local magnetic susceptibilities in the Hubbard model and find a finite anomalous term in the Mott insulating regime corresponding to non-decaying long-term magnetic correlations.

The article is well written, and the results are interesting. Furthermore, this analysis can be implemented for different models and correlation functions. I support the publication of this manuscript.

Before publication, I would ask the authors for a few changes listed below.

Requested changes

(1) While the thermal and real-time susceptibilities have been defined using equations, I could not find the definitions of chi^T and chi^S. These definitions should be included.

(2) While reading this manuscript, I was not sure which quantities actually have been calculated with DMFT/QMC and which quantities have been obtained by analytic continuation or are used as parameters.
For example, how was A(w) obtained in equation (44)? Was it obtained by analytical continuation from the same data set as chi^th?
In equation (45), chi^R has been obtained by analytical continuation?
It should be clarified how these quantities have been obtained!

(3) If analytical continuation has been used to calculate chi^R from which C is calculated, how large is the error on C originating in an analytical continuation?

(4) Concerning the anomalous term at finite temperatures, is there an exact expression for large temperatures? If so, it would be good to compare to this in Fig. 7.

(5) In equation (44), are K, g_b, and sigma the same functions as used in the analytical continuation section?

(6) What happened with the delta function when going from equation 16 to 17?

(7) The gray colors in the right top panel in Fig. 1 are hard to see. Maybe the authors can change this color?

(8) In Fig. 3, there are only three curves visible but the legend includes 4 parameters.

  • validity: high
  • significance: high
  • originality: high
  • clarity: good
  • formatting: excellent
  • grammar: excellent

Author:  Clemens Watzenböck  on 2022-04-05  [id 2357]

(in reply to Report 2 on 2022-01-12)
Category:
answer to question

We thank the Referee for her/his positive assessment on both our results and presentation, and for supporting publication in SciPost Physics. In particular, we appreciate the constructive criticisms of the Referee aiming at improving the clarity of selected paragraphs.
In the resubmitted manuscript we have taken care of all the points raised in her/his report. A structured reply to the observations of the Referee is enclosed below:

  1. While the thermal and real-time susceptibilities have been defined using equations, I could not find the definitions of $\chi^T$ and $\chi^S$. These definitions should be included.

    Thank you for pointing this out. To make the paper self-contained we have now included the definitions in Eq. (2).

  2. While reading this manuscript, I was not sure which quantities actually have been calculated with DMFT/QMC and which quantities have been obtained by analytic continuation or are used as parameters. For example, how was $A(\omega)$ obtained in equation (44)? Was it obtained by analytical continuation from the same data set as $\chi^{\mathrm{th}}$? In equation (45), $\chi^R$ has been obtained by analytical continuation? It should be clarified how these quantities have been obtained!

    In order to avoid possible misunderstandings, we now state explicitly in section 3.2 that all quantities in real frequencies shown in our paper are obtained by analytic continuation. We have also added some more specific statement by discussing Eqs. (44)-(45). [Eqs. (45)-(46) in the new version; see provided latexdiff].

  3. If analytical continuation has been used to calculate $\chi^R$ from which $C$ is calculated, how large is the error on $C$ originating in an analytical continuation?

    The error-estimate strongly depends on the signal-to-noise ratio and the temperature. On the one hand, by increasing $T$ the first-Matsubara frequency will be further away from the zeroth one. Hence, one has to extrapolate over a larger frequency distance. This problem can – at least in part – be mitigated by demanding a temperature-dependent minimal blurwidth. On the other hand, the difference between the isothermal and Kubo susceptibility $\chi^{\mathrm{th}}(\mathrm{i} \omega_n=0) - \chi^{\mathcal{R}}(\omega=0) = \beta C$ becomes smaller for larger temperatures. This will lead to a larger error on our estimate of $C$ in the high-temperature regime.

    Motivated by the Referee's observation, we have now included an additional analysis in the appendix for test data that is similar to measured QMC-data. Our analysis shows that for $U=3$ the error-estimate is $<1\%$ for low temperatures and for the highest considered temperature still $<5\%$. inside of the coexistence region just before metal-to-insulator phase transition we estimate the error to be rather large ($C=0.06 \pm 0.05$). The intrinsic difficulty there is that we try to distinguish a very sharp peak at finite frequency (preformed local moment with a finite life-time) from a formally infinitely sharp peak at $\omega=0$ (anomalous term). In any case, our error estimate is also consistent with the right part of Fig. 5. In particular, after a careful analysis of the fit-loss, as outlined in Eq. (44), we have concluded that $C\approx 0$ is consistent with our QMC data.

    For the sake of clarity, we now mention explicitly in the main text that our estimated value for $C$, which is finite though small, should be regarded as an artifact of the extraction method, which results from a bad signal-to-noise ratio. We have also included a simple error analysis for some selected cases in appendix D.

  4. Concerning the anomalous term at finite temperatures, is there an exact expression for large temperatures? If so, it would be good to compare to this in Fig. 7.

    In response to this Referee's question, we have now included a small section in the appendix E, where we explicitly address the large-temperature limit ($T \gg U, W$). A direct comparison with Fig. 7 is, however, not possible, because even the highest temperature ($T=0.1$) reported there, it is still very low compared to the other relevant energy scales of the problem (like $U$ or the bandwidth $W$). We now elaborate on this point in the new appendix section.

  5. In equation (44), are $K$, $g_b$, and $\sigma$ the same functions as used in the analytical continuation section?

    Yes, they are. We added a sentence below Eq. (44) to clarify this.

  6. What happened with the delta function when going from equation 16 to 17?

    We used that for $\chi^c(\omega)=-\mathrm{i}(\mathrm{e}^{\beta \omega} - 1)\chi^<(\omega)$ the prefactor $(\mathrm{e}^{\beta \omega} - 1)$ is zero for $\omega=0$. We have now added a specific footnote to make this step clearer.

  7. The gray colors in the right top panel in Fig. 1 are hard to see. Maybe the authors can change this color?

    Thank you for pointing this out. Indeed, we verified that in the printout of one of us the lines were hardly visible, too. In order to avoid possible print-setting related problems, we have modified the colors in Fig.~1 accordingly.

  8. In Fig. 3, there are only three curves visible but the legend includes 4 parameters.

    There is also a fourth curve that is almost on top of another one. We now modified some of the plot-markers to increase visibility.

Attachment:

reply_and_latexdiff_LTM.pdf

---

## Round 2 · Referee Report · Andrey Katanin (Referee 1) · 2022-4-19

Strengths

The manuscript was substantially improved (despite even the first iteration was already of high quality), the authors considered all questions and comments of the Referees.

Weaknesses

-

Report

The authors considered all questions and comments of the Referees and introduced the necessary changes. The readability of the paper has substantially improved, some important new aspects (e.g. approximate analytic forms of chi^R, together with the corresponding comparison to the oscillator form, high temperature limit, error estimate, etc.) were added.

Regarding the new Table 1 of Appendix C, I consider it is interesting that: (i) w0<gamma, i.e. the oscillators, which are considered by the authors, are over damped, and (ii) the difference between the obtained w0 and gamma is rather small, such that the two poles of \chi^R(w), discussed by the authors in Appendix C, are nearly degenerate. This might imply the possibility of using the form, which is a modified version of Ref. [61] of the paper, chi^R(w)=-A delta^2/(w+i delta)^2, in future studies.

I recommend the revised paper for publication.

Requested changes

-

---

## Round 2 · Referee Report · Anonymous (Referee 2) · 2022-4-20

Report

The authors have satisfactorily answered all questions giving many details.
This work presents excellent research on strongly correlated electron systems.
I recommend it for publication in Scipost Physics.

---

## Round 2 · Author Response

Dear Editor,

Thank you very much for forwarding the two referee reports on our
manuscript 2112.02903v1

Long-term memory magnetic correlations in the Hubbard model: A dynamical mean-field theory analysis.

We thank Referee A (Prof. A. Katanin) and Referee B for their very positive judgment on the quality
and the impact of our work, which they find *"interesting"* and *"well written"*.
We also thank the referees for stating that our work *"sheds new light"* on the subject and that it allows
for a *"deeper understanding"*.

While both Referees consider our work suited for publication in SciPost Physics,
they have added specific and constructive suggestions in their reports,
aiming at improving the impact and the clarity of our presentation.
We have carefully considered all of them in our resubmitted manuscript.
A list of the corresponding changes is enclosed below.

We hope that our manuscript, incorporating all the helpful suggestions of both referees,
can now be accepted for publication in SciPost Physics.
The new version 2112.02903v2 is now available on arXiv.

Thank you very much for your assistance,

Sincerely yours,

C. Watzenböck, M. Fellinger, K. Held and A. Toschi

---

## Round 2 · List of Changes

Warnings issued while processing user-supplied markup:

  • Inconsistency: Markdown and reStructuredText syntaxes are mixed. Markdown will be used.
    Add "#coerce:reST" or "#coerce:plain" as the first line of your text to force reStructuredText or no markup.
    You may also contact the helpdesk if the formatting is incorrect and you are unable to edit your text.

REPLY TO REFEREE A

We thank the Referee for the careful reading of our manuscript, for his appreciation of our work and for finding our results suited for being presented in SciPost Physics. The Referee has made constructive and useful comments/observations in his report. They have helped us improving the clarity and the precision of our presentation as well as the completeness of our bibliography. A structured reply to the observations of the Referee is enclosed below:

  1. The authors use a notion of isolated (Kubo) susceptibility, which is also called by them sometimes isolated (Kubo-Nakano) susceptibility in the abstract and main text of the paper. I am not sure that this notion is rather common (in particular, in what sense the susceptibility (or system) is isolated?). I would suggest to name this quantity as a static limit of the retarded spin correlation function, or somehow else.

    We thank the Referee for his observation. In the revised version of the manuscript, we have uniformed the choice of lexicon, privileging the definitions most commonly used in the literature and trying, at the same time, to convey more precisely the essential information to the readership. This is particularly important here, because of the central role played by the difference between the two definitions of susceptibility in the problem of long-term memory effects.

    In a nutshell, the first susceptibility $\chi^{\mathcal{R}}(\omega)$, which corresponds to the standard definition within the Kubo-Nakano theory of linear response, describes how a many-electron systems system reacts to an external dynamical perturbation ${\cal F}(t)$ applied after a certain time $t=t_0$. It is important to notice here that, before the perturbation is applied, the system is in thermal equilibrium with an external bath at an (inverse) temperature $\beta$. When subsequently computing its response to the applied perturbation for $t>t_0$, one assumes that no further thermalization process occurs (cf. Eq. (1), where the grand-canonical summation entails the Boltzmann factors of the unperturbed Hamiltonian). In other words, the system is considered isolated in the course of the measurement. The second relevant case (The isentropic susceptibility $\chi^S$, also briefly mentioned in Sec. II, does not play a particularly significant role in our study.) is represented by the static isothermal susceptibility $\chi^T$, and corresponds -- at a certain extent -- to the opposite situation, where one applies a completely static perturbation (${\cal F}=$ const. at all times) and allows for a full thermalization of the perturbed system with the external bath [cf. Eq. (2)].

    According to Eqs. (5-6) the long-term memory effects, encoded by the term $\beta C$, can be computed from the difference between the static isothermal response $\chi^T$ and the zero-frequency limit of the Kubo dynamical response $\chi^{\mathcal{R}}(\omega \rightarrow 0)$. In fact, the possibility of finding a finite difference between these two quantities (even when the limit $\omega \rightarrow 0$ is taken for the latter) and, hence, of observing a non-decaying behavior of the corresponding correlations in time, directly originates from the intrinsic difference between the specific "experimental" set-ups (i.e., isothermal vs. isolated) of the two susceptibilities considered.

    On the basis of the Referee's remark and the above considerations, we have adopted throughout the manuscript (including the abstract) the following terminology: We refer to $\chi^T$ as (static) isothermal susceptibility and $\chi^{\mathcal{R}}(\omega \rightarrow 0)$ as (the zero frequency limit of) the Kubo susceptibility. In this perspective, we explicitly highlight the important relation of the latter with a perturbation acting on an (otherwise) isolated system at the beginning of Sec. II, whose wording has been accordingly extended and improved, but, as suggested by the Referee. We avoid the less usual terminology of "isolated susceptibility" in the rest of the manuscript.

  2. In the second and third paragraph of the Introduction I think it is important to mention everywhere that the authors talk about slowing-down of local spin fluctuations ("local" is mentioned only once in the beginning of second paragraph, but it is preferable I think to insert it further, since this is crucial to differentiate this from the critical slowing down etc.).

    We have inserted the specification about the locality of the spin susceptibility considered in three points of the Introduction, where its inclusion was highly pertinent. However, we kept the more general term whenever we refer to generic properties of the response theory. This has made the presentation of our work more precise and easily understandable for the reader. We thank the Referee for the suggestion.

  3. Regarding this "local" slowing down, I think it is important to cite the paper 10.1103/PhysRevLett.101.166405 which was one of the first papers which discussed this topic for Hund metals.

    We thank the referee for pointing this out to us. The inclusion of this reference is highly pertinent. We now cite this work in two places (see latexdiff below). In this respect, it is worth to remark that their "spin-freezing" result is related to the appearance of the anomalous ("long-term-memory") effect ($C>0$) rather than to the mere slowing of the spin-dynamics. At the same time, other works of the same group have also analyzed the slowing down effects in the dynamics of orbital fluctuations (for multiorbital systems with effective attractive interaction). For the sake of completeness we also included references to these results in the first part of the Introduction.

  4. Regarding Eqs. (25), (28), as well as Eqs. (19), (20), I would like to emphasize that the "anomalous" C-terms can be viewed as a certain limit of some regular contribution to the susceptibility. One such phenomenological form of the regular part, which provides these anomalous terms, was suggested in Eq. (2) of Ref. 10.1103/PhysRevB.88.155120, and reads in the notations of the author's paper \begin{equation} \chi^R(\omega)=C\beta i\delta/(\omega+i\delta), \quad \text{where}\quad \delta \rightarrow +0 \tag{1} \end{equation} Then \begin{align} \Im(\chi^R(\omega)) & = C\beta \omega \, \delta/(\omega^2+\delta^2) \tag{2} \ \Re(\chi^R(\omega)) & = C\beta \delta^2/(\omega^2+\delta^2) \tag{3} \ \chi^{th}(i\omega_n) & = C\beta\delta/(\omega_n+\delta) \rightarrow C \beta \delta_{n,0} \quad \text{ at } \quad \delta \rightarrow 0. \tag{4} \end{align}

    Indeed, the Referee statement that the anomalous term $\propto \beta C$ can be also captured by taking a specific limit of formally regular spectral expression, namely by considering $\delta \rightarrow 0^{+}$ in the equations reported above, is correct.

    In particular, this procedure would lead in Eq. (19) and (20) to $\chi^c(\omega) \rightarrow \chi^c(\omega) + 2 \pi C \beta \omega \, \delta(\omega)$ and in Eq. 28 to $\Im \chi^{\mathcal{R}}(\omega) \rightarrow \Im \chi^{\mathcal{R}}(\omega) + \pi C \beta \, \omega \, \delta(\omega) $. Hence, by following this alternative route, one could eventually re-derive the same analytical expressions we reported in our manuscript.

    At the same time, it should be stressed that the intrinsic "anomalous" nature of the zero-frequency contribution of the above equations appears reflected, after the limit $\delta \rightarrow 0^{+}$ is taken, in the peculiar expression of the corresponding $\delta$-singularity with vanishing weight $ \propto C \beta \omega \, \delta(\omega)$ in $\Im(\chi^R(\omega))$.

    It is also important to notice, both for practical application and for a fundamental understanding, that the order in which the limits are evaluated plays a crucial role here. For instance, if one considers the (inverse) Fourier-transform of Eq. (1)

    $$ \chi^R(t, \delta) = C \, \beta \, \theta(t) \, \delta \, \mathrm{e}^{-\delta \cdot t}, $$

    which encodes relevant information about the dynamical properties directly on the real time axis, and then takes the limit $\delta \rightarrow 0^{+}$, one gets:

    $$ \lim_{\delta \rightarrow \, 0^{\, +}} \, \chi^R(t,\delta) = \lim_{\delta \rightarrow \, 0^{\, +}} \, C \, \beta \, \theta(t) \; \delta \; \mathrm{e}^{-\delta \cdot t} \equiv 0. $$

    Such a formally vanishing result would then correspond, after Fourier transforming back to real frequencies, to an exactly zero spectral function everywhere and, hence, to an apparently inconsistent outcome w.r.t. the expressions discussed above. In fact, the intrinsic delicateness of this procedure is a direct consequence of the fact that, as we discuss at the beginning of Sec. II of our manuscript, the Kubo susceptibility, corresponding to the (retarded) commutator contribution $\chi^c(t)$, does not necessarily entail the full information about the static response of the system. This aspect can be more directly addressed by considering that the inclusion/exclusion of the peculiar singular term discussed above, i.e., $\Im \chi^R(\omega)\propto\omega\, \delta(\omega)$ can only yield a zero-measure extra contribution in the real part of the response function, namely :

    $$ \Re \chi^R(\omega) = \bigg{ \begin{array}{lr} C\beta & \text{for } \omega=0\ 0 & \text{otherwise } \end{array} $$

    More formally, this corresponds to saying that the presence such an additional term generated by the $\delta = 0^{+}$ limit of Eq. (1) can only affect the anti-commutator contribution, which fully encodes the static isothermal (i.e., non Kubo) response:

    $$ \Psi(\omega)= 2 \pi C\beta \delta(\omega) \omega \coth(\beta/2\omega) = \mathrm{i} 4 \pi \delta(\omega) C. $$

    As a consequence, also in Eq. (4), it is crucial in which order the limits are taken:

    $$ \begin{array}{ccccl} \lim\limits_{\delta \rightarrow 0 } & \lim\limits_{\mathrm{i} \omega_n \rightarrow z \rightarrow 0 } & \chi^{\mathrm{th}}(\mathrm{i} \omega_n) &=& \beta C \quad [\mbox{isothermal}]\ \lim\limits_{\mathrm{i} \omega_n \rightarrow z \rightarrow 0 } & \lim\limits_{\delta \rightarrow 0 } & \chi^{\mathrm{th}}(\mathrm{i} \omega_n) &=& 0 \qquad [\mbox{Kubo}] \ \end{array} $$

    In conclusion, we do agree with the Referee that the anomalous part of the response, responsible for the long-term memory effects described in our manuscript could also be extracted by performing specifically defined limit-operations on formally regular spectral contributions. However, after the limits (rigorously following the correct order) are taken, the resulting zero-frequency term becomes indeed singular and formally
    corresponds to the anomalous term described in our manuscript. As the latter is defined by the difference between the static/isothermal and the zero-frequency limit of the Kubo response (cf. Eq. (5) in our manuscript), it is not surprising that the way in which the limits are performed is crucial to correctly capture the possible presence of an anomalous contribution to the static response.

  5. It would be good in my opinion to clarify how the first order transition line in Fig. 1 is determined. Is it obtained from the comparison of full energies of metallic and insulating solutions, or their free energies, or something else?

    Yes, the Referee is right. The thermodynamical transition line has been obtained (in Blümer (2002); Ref. [23] in version 1) through the comparison of the respective free energies. Indeed, in the previous version of our manuscript this information was only reported in the caption of Fig. 5. In consideration of the Referee's remark, we have now added it also to the main text in the proximity of Fig. 1.

  6. The definition of the Widom line is in my opinion better present in the beginning of Sect. 4 (first line of page 14).

    Thanks for the suggestion. We moved the definition of the Widom line from page 20 to page 14. See attached latexdiff.

  7. In Sect. 4.1 the authors discuss "rapid decay" of local spin correlation function in the insulating phase; I think it is better to characterise it as a rapid decrease, since change in the magnitude of the correlation function is not large in that regime.

    Indeed, we write in Sec. 4.1 about "a rapid decay to rather large constant offset" and "significantly slower decay towards ...". In this respect, we think that our formulation already prevents possible misunderstandings, because we have explicitly mentioned to which quantity the correlation function is "decaying" (i.e., here "to a rather large constant offset"). Since the latter is a (large) constant offset, the word "decay" appears pertinent in this context.

  8. For the data presented in the right part of Fig. 2 and left part of Fig. 4 the authors might consider the comparison to the analytic forms discussed in point 4) above and/or suggest their own analytic forms.

    As the Referee explicitly mentions the left part of Fig. 4, we assume that he is referring to the regular part of the response function. In this respect the particular form discussed at point 4) is not suitable. The reason is the following: Any Kubo susceptibility where $\hat A = \hat B$ is zero – per construction – for $t=0^+$ since $\chi^{\mathcal{R}}_{AA}(t\rightarrow0^+) \propto \langle[\hat A(t\rightarrow0^+), \hat A]\rangle=0^+$. The suggested form at point 4) above, instead, has as a Fourier-transform $\chi^{\mathcal{R}}_L(t) \propto \theta(t) \mathrm{e}^{-\delta \cdot t}$. Hence, it does not fulfill this boundary condition of a Kubo response for a finite $\delta$.

    Consistent with this consideration, when we try to apply it to our data sets, it does not yield a good fit of the data.

    In order to fulfill the boundary condition $\chi^{\mathcal{R}}(\omega)$ must display at least two (simple) poles in the lower complex half plane.

    As a simple analytic function which does fulfill the boundary condition mentioned above, we suggest to consider a function with two poles at $\Omega_\pm = -\mathrm{i}\gamma \pm \sqrt{\omega_0^2 - \gamma^2}$, which corresponds to the case of a damped harmonic oscillator. In the revised manuscript, we have included fits to the QMC data for selected cases of Fig. 2 and compared it to Fig. 4 in the appendix C.

  9. Regarding right part of Fig. 5: do I understand correctly that finite (although small) C in the metallic phase (M$\rightarrow$I path) is an artefact of the numerical procedure of calculating C? If yes, I think it is worth to write this explicitly.

Yes, this is correct. Following the Referee suggestion, we have improved the corresponding sentence in the main text, and also included an error-estimate on $C$ for selected cases in Appendix D.

  1. In the upper plot of the right part of Fig. 7 I think it is worth to mark the position of the Widom line.

    Thanks for the suggestion. In the revised manuscript, we have added a dashed blue line marking the Widom line to the upper right part of Fig. 7.

REPLY TO REFEREE B

We thank the Referee for her/his positive assessment on both our results and presentation, and for supporting publication in SciPost Physics. In particular, we appreciate the constructive criticisms of the Referee aiming at improving the clarity of selected paragraphs.
In the resubmitted manuscript we have taken care of all the points raised in her/his report. A structured reply to the observations of the Referee is enclosed below:

  1. While the thermal and real-time susceptibilities have been defined using equations, I could not find the definitions of $\chi^T$ and $\chi^S$. These definitions should be included.

    Thank you for pointing this out. To make the paper self-contained we have now included the definitions in Eq. (2).

  2. While reading this manuscript, I was not sure which quantities actually have been calculated with DMFT/QMC and which quantities have been obtained by analytic continuation or are used as parameters. For example, how was $A(\omega)$ obtained in equation (44)? Was it obtained by analytical continuation from the same data set as $\chi^{\mathrm{th}}$? In equation (45), $\chi^R$ has been obtained by analytical continuation? It should be clarified how these quantities have been obtained!

    In order to avoid possible misunderstandings, we now state explicitly in section 3.2 that all quantities in real frequencies shown in our paper are obtained by analytic continuation. We have also added some more specific statement by discussing Eqs. (44)-(45). [Eqs. (45)-(46) in the new version; see provided latexdiff].

  3. If analytical continuation has been used to calculate $\chi^R$ from which $C$ is calculated, how large is the error on $C$ originating in an analytical continuation?

    The error-estimate strongly depends on the signal-to-noise ratio and the temperature. On the one hand, by increasing $T$ the first-Matsubara frequency will be further away from the zeroth one. Hence, one has to extrapolate over a larger frequency distance. This problem can – at least in part – be mitigated by demanding a temperature-dependent minimal blurwidth. On the other hand, the difference between the isothermal and Kubo susceptibility $\chi^{\mathrm{th}}(\mathrm{i} \omega_n=0) - \chi^{\mathcal{R}}(\omega=0) = \beta C$ becomes smaller for larger temperatures. This will lead to a larger error on our estimate of $C$ in the high-temperature regime.

    Motivated by the Referee's observation, we have now included an additional analysis in the appendix for test data that is similar to measured QMC-data. Our analysis shows that for $U=3$ the error-estimate is $<1\%$ for low temperatures and for the highest considered temperature still $<5\%$. inside of the coexistence region just before metal-to-insulator phase transition we estimate the error to be rather large ($C=0.06 \pm 0.05$). The intrinsic difficulty there is that we try to distinguish a very sharp peak at finite frequency (preformed local moment with a finite life-time) from a formally infinitely sharp peak at $\omega=0$ (anomalous term). In any case, our error estimate is also consistent with the right part of Fig. 5. In particular, after a careful analysis of the fit-loss, as outlined in Eq. (44), we have concluded that $C\approx 0$ is consistent with our QMC data.

    For the sake of clarity, we now mention explicitly in the main text that our estimated value for $C$, which is finite though small, should be regarded as an artifact of the extraction method, which results from a bad signal-to-noise ratio. We have also included a simple error analysis for some selected cases in appendix D.

  4. Concerning the anomalous term at finite temperatures, is there an exact expression for large temperatures? If so, it would be good to compare to this in Fig. 7.

    In response to this Referee's question, we have now included a small section in the appendix E, where we explicitly address the large-temperature limit ($T \gg U, W$). A direct comparison with Fig. 7 is, however, not possible, because even the highest temperature ($T=0.1$) reported there, it is still very low compared to the other relevant energy scales of the problem (like $U$ or the bandwidth $W$). We now elaborate on this point in the new appendix section.

  5. In equation (44), are $K$, $g_b$, and $\sigma$ the same functions as used in the analytical continuation section?

    Yes, they are. We added a sentence below Eq. (44) to clarify this.

  6. What happened with the delta function when going from equation 16 to 17?

    We used that for $\chi^c(\omega)=-\mathrm{i}(\mathrm{e}^{\beta \omega} - 1)\chi^<(\omega)$ the prefactor $(\mathrm{e}^{\beta \omega} - 1)$ is zero for $\omega=0$. We have now added a specific footnote to make this step clearer.

  7. The gray colors in the right top panel in Fig. 1 are hard to see. Maybe the authors can change this color?

    Thank you for pointing this out. Indeed, we verified that in the printout of one of us the lines were hardly visible, too. In order to avoid possible print-setting related problems, we have modified the colors in Fig.~1 accordingly.

  8. In Fig. 3, there are only three curves visible but the legend includes 4 parameters.

    There is also a fourth curve that is almost on top of another one. We now modified some of the plot-markers to increase visibility.

LIST OF CHANGES

All changes are marked in blue/red in the manuscipt (latexdiff) attached to this reply. Besides the changes already discussed in answer to referees above, the following changes have been made:

  1. There was an error in Eq. (3) and Eq. (5) ($\langle \hat A(t)\hat B\rangle$ instead of $\langle {\hat B \hat A(t)} \rangle$) and in Eq. (34) ($\hat n_{i\uparrow} \hat n_{j\downarrow}$ instead of $\hat n_{i\uparrow} \hat n_{i\downarrow}$).

  2. We forgot a factor of $\pi$ when calculating $\Im \chi^{\mathcal{R}}(\omega)$ from the MaxEnt output. We now rescaled $\Im \chi^{\mathcal{R}}(\omega)$ in Fig. 4 and 7 to get the correct result.

  3. We fixed some references (added first name of author for Ref. 52 and Ref. 53.; added arXiv link for Ref. 48).

---

## Editorial Decision

published